# Triple-interlocked-nanotwinned bulk magnesium alloys with exceptional strength and ageing resistance

Qiuming Peng [1,6] ✉, Lutong Zhou[1,6], Jinming Wang [1,6], Ke Tong[1], Wentao Hu[1], Lin Wang[1], Yong Sun[1], Yipeng Gao[2], Anmin Nie[1], Biaobiao Yang[3,4], Wei Cai[5], Guodong Zou[1] ✉, Tianlin Huang[5] ✉, Wanquan Zhu[5] & Yongjun Tian [1] ✉

Nanoscale twin boundaries (TBs) effectively restrict the free motion of dislocations by intersecting with other TBs to realize high strength. However, the dislocations parallel to the TBs can glide along the TBs, resulting in detwinning for low strength. Herein, we present a triple-interlocked-nanotwinned (TIT) strategy for overcoming these inherent deficiencies. An Mg-9Li alloy was used as a proof-of-concept. The treated alloy had a large volume fraction of TIT interfaces (65.7%). It not only had a high yield strength of 508.6 MPa (approximately eleven times higher than that of the TIT-free sample) but also offered unprecedented ageing resistance, with a reduction in hardness of 6% after 730 days. The martensite-like phase transformation process concurrently triggers multiple-variant twins to form TIT structures. These TIT nets can effectively prohibit dislocation motion, reduce strengthening anisotropy, and disperse localized stress during the deformation process. This strategy of phase transformation twinning open a gate to fabricate other stronger metallic or ceramic nanotwinned bulk materials.

High-performance structural metals strongly affect socioeconomic development and are especially desirable for pursuing sustainable human development and carbon neutrality targets. For all serviced structural metals, the development of next-generation alternatives with higher strength and better ductility become a common issue[1]. According to classical dislocation theory, the main point lies in tailoring the balance between dislocation pinning and dislocation motion by means of interface barriers, wherein dislocation motion can be effectively prohibited or properly released. High strength in metallic alloys is typically attained through three primary strengthening mechanisms (either individually or in combination): refined grain strengthening, precipitation strengthening, and solid-solution strengthening.

Specifically, the decrease of grain size can remarkably increase the volume fraction of grain boundaries (GBs), wherein they act as effective obstacles to prohibit dislocation motion for high strength. Meanwhile, the accumulated dislocations in GBs result in stress concentration during the deformation process for low ductility[2]. Comparatively, in the cases of precipitation strengthening and solid-solution strengthening, it is easy to form some incoherent or semi-coherent phase or cluster interfaces[3]. Due to the different modulus between phase or cluster and matrix, the dislocation motion can be well inhibited, thereby enhancing its strength. More importantly, they also increase strain localization in the interface and then cause early failure, leading to an unavoidable trade-off between strength and ductility[4].

[1]State Key Laboratory of Metastable Materials Science and Technology, Yanshan University, Qinhuangdao, P. R. China. [2]Key Laboratory of Automobile Materials of Ministry of Education & School of Materials Science and Engineering, Nanling Campus, Jilin University, Changchun, P. R. China. [3]IMDEA Materials Institute, Madrid, Spain. [4]Department of Materials Science, Polytechnic University of Madrid, Universidad Politécnica de Madrid, E.T.S. de Ingenieros de Caminos, Madrid, Spain. [5]International Joint Laboratory for Light Alloys, College of Materials Science and Engineering, Chongqing University, Chongqing, P. R. China. [6]These authors contributed equally: Qiuming Peng, Lutong Zhou, Jinming Wang. ✉e-mail: pengqiuming@ysu.edu.cn; zouguodong@ysu.edu.cn; huangtl@cqu.edu.cn; fhcl@ysu.edu.cn

Like refined grain strengthening, the introduction of nanoscale twin boundaries (TBs) enhances strength by reducing the mean free path of dislocations[5]. Moreover, low-energy coherent TBs contribute to improved ductility by enabling dislocation transmission and storage along TBs[6], which facilitates strain accommodation and delays crack initiation. Thus, twinning strengthening has been introduced as a universal path for the pursuit of superior mechanical and physical properties. For example, electrodeposited nanotwinned copper[7], electrodeposited nanotwinned nickel[8] and cryoforged nanotwinned titanium[9] have displayed exceptional synergy between strength and ductility. Additionally, this twin strategy is effective even in cubic boron nitride[10] and diamond[11], thus pushing hardness to new records. Conventional twin strategies are associated mainly with a single twin variant, in which a high TB interface fraction is achieved by reducing the twin thickness. In this context, there are two inherent deficiencies in traditional twin engineering. Specifically, the deformation perpendicular to the TBs is effectively hindered, whereas the strengthening role parallel to the TBs weakens because of dislocation glide, thereby leading to increased anisotropy[12]. Moreover, when the twin thickness is below a critical value, a detwinning phenomenon that is caused by the nucleation and motion of partial dislocations parallel to the TBs occurs[13].

Herein, we proposed a structural design strategy for overcoming these intrinsic deficiencies of twin engineering. Departing from traditional GBs and phase interfaces, a unique triple-interlocked-nanotwinned (TIT) structure is developed (Fig. 1a–c). Namely, we simultaneously activate new nanotwins with various orientations in the whole matrix, and then these twin variants generate TIT structures during their growth by optimizing intersection angles ($\theta$) and twin interfaces ($t$) through a solid-state phase transformation under special stress–strain conditions of high pressure and high temperature (HPHT). Noticeably, in contrast to growth-twinning and deformation-twinning, which have been used to generate nanotwinned structures in pure metals and ceramics, solid-state phase transformation twinning eliminates the hierarchical strain distribution[14], and then breaks through the dimension limitations of nanotwinned bulk materials. Theoretically, the TIT structure can prohibit dislocation motion along three different glide planes (Fig. 1d), thereby effectively reducing dislocation motion disparities. Simultaneously, the formation of steps in TBs might enable high ductility during the deformation process. Consequently, compared with other interface strengthening mechanisms (Fig. 1e), the TIT structure not only impedes twin widening but also weakens the strengthening anisotropy. More attractively, a higher twin interface fraction can be obtained with the same twin spacing, thus possibly eliminating the inverse Hall–Petch relationship by preventing partial dislocations gliding or by inhibiting detwinning.

## Results and discussion
### Unique TIT structure
To validate this design concept, we chose a simple Mg-9Li (wt.%) alloy as a model on the basis of the following two factors. Firstly, in this typical hexagon-closed-packed (HCP) alloy, twinning commonly occurs to coordinate Mg matrix deformation owing to the lack of

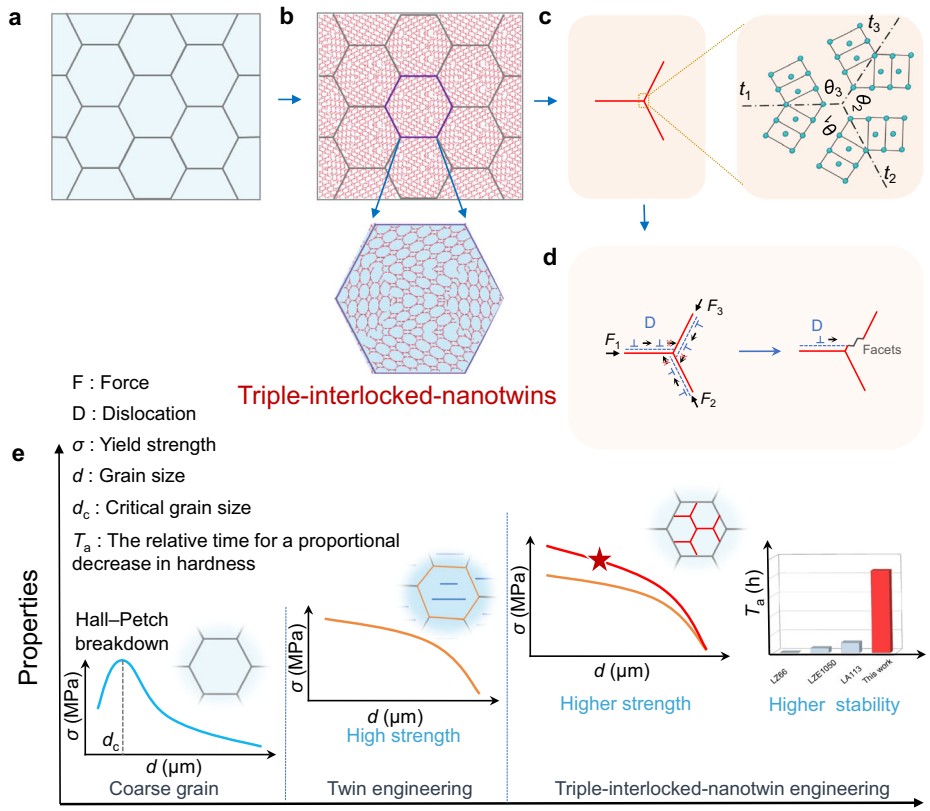

**Fig. 1 | Schematic diagram of triple-interlocked-nanotwinned concept.**
**a** Coarse-grained microstructure. **b** Design concept of the TIT structure. **c** Atomic illustration of a TIT unit, where $t_1$, $t_2$, and $t_3$ represent the twin interfaces in three different directions and $\theta_1$, $\theta_2$, and $\theta_3$ correspond to the angles between two adjacent TBs. **d** The interaction between TBs and dislocations during the deformation process. **e** Schematic representation of the effects of TIT structures on mechanical properties. Unlike coarse-grained strengthening materials, nanograined materials exhibit Hall–Petch breakdown phenomena below a critical grain size ($d_c$). Continuous hardening phenomena are detected in twin engineering with decreasing twin thickness, yet the typical single twin variant results in increased anisotropy. TIT structures increase strength and structural stability ($T_a$: the relative time for a proportional decrease in hardness, the detailed data comparison is shown in Fig. 5d) by overcoming the inherent deficiencies of conventional twin strengthening.

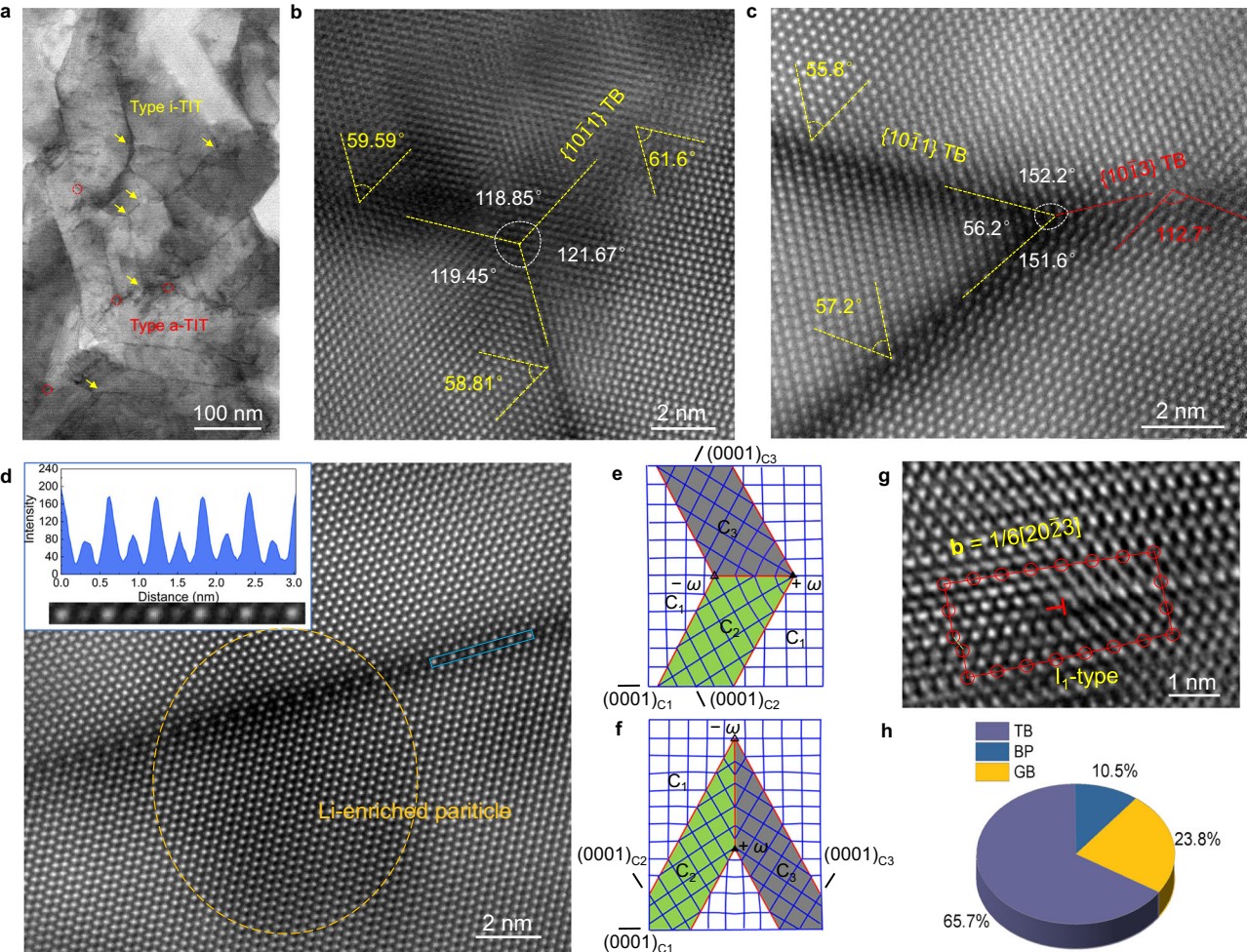

**Fig. 2 | Microstructures. a** BF-STEM image of TIT structures in the HPHT-800 Mg-9Li alloy viewed along [11$\bar{2}$0]. **b** HAADF-STEM image of a typical i-TIT structure. **c** HAADF-STEM image of a typical a-TIT structure. The yellow dashed lines and red dashed lines represent {10$\bar{1}$1} TBs and {10$\bar{1}$3} TBs, respectively. **d** HAADF-STEM image of a typical {10$\bar{1}$1} TB. Li atoms are arranged periodically on the {10$\bar{1}$1} TB. **e, f** The disclination structures of i-TIT and a-TIT, respectively, where ω represents

the disclination and $C_1$–$C_3$ denote the different twin variants. The blue solid line represents the Mg matrix plane, and the red solid line represents the {10$\bar{1}$1} TBs. **g** HRTEM images of basal intrinsic $I_1$-type stacking faults around {10$\bar{1}$1} TBs viewed along [11$\bar{2}$0]. **h** The statistical fractions, determined on the basis of TEM images, of various interfaces. Source data for (**d**) and (**h**) are provided as a Source Data file.

enough independent slip systems along the c-axis. Additionally, the Mg-9Li alloy exhibits a duplex structure: an HCP-type Mg-rich phase (α-phase) and a body-centred cubic (BCC) Li-rich phase (β-phase). Furthermore, the HCP or BCC phase fraction can be manipulated by adjusting the pressure and temperature[15]. Therefore, the Mg-9Li alloy provides suitable conditions for simultaneously triggering different twin variants to form a TIT structure under HPHT conditions.

We performed a two-step HPHT treatment with an industrial cubic-anvil large-volume press with six rams on the as-cast Mg-9Li alloy (with a diameter and height of 30 and 15 mm, respectively; Supplementary Fig. 1). A representative dendrite structure changed into an equiaxed grain with a grain size of hundreds of micrometres after the HPHT treatment. As shown by the bright-field transmission electron microscopy (BF-STEM) image (Fig. 2a and Supplementary Fig. 1), TIT structures were present in the HPHT-800 Mg-9Li alloy (1 GPa at 400 °C followed by 6 GPa at 800 °C). The statistical twin thickness was 47 ± 10 nm (Supplementary Fig. 2). According to the local high-resolution high-angle annular dark-field (HAADF) images, the TIT structure was composed of two main substructures: isometric triple-interlocked nanotwins (i-TITs) and anisotropic triple-interlocked nanotwins (a-TITs). Further structural analysis revealed that the

i-TITs were composed of three {10$\bar{1}$1} contraction twin interfaces, as confirmed by a measured rotation angle of 58.81–61.6° between the matrix and twin planes[16] (Fig. 2b). Comparatively, for the a-TIT structure, the two interfaces consisted of {10$\bar{1}$1} contraction twins with an intersection angle of 55.8 - 57.2°[17], whereas the third interface was composed of a {10$\bar{1}$3} twin with an angle of approximately 112.7° between the matrix and twin plane[18] (Fig. 2c). The phase relationship is further confirmed by SAED patterns corresponding to the a-TIT structure and the i-TIT structure (Supplementary Fig. 3). Moreover, similar to other rare earth elements[18], Li atoms are periodically arranged at the {10$\bar{1}$1} contraction twin interfaces (Fig. 2d).

To understand the spatial characteristics of the TIT structure, we adopted disclination to explore the formation of triple junctions (Supplementary Note 1). $C_1$ represents the undeformed matrix, and $C_2$ and $C_3$ denote two states of contraction twinning in the Mg alloy. As illustrated in Fig. 2e, f, Supplementary Fig. 4, and Supplementary Table 2, the $(1\bar{1}0\bar{1})_{C1}$ and $(1\bar{1}01)_{C1}$ twins intersected under specific loading conditions, and a new twin plane $(1\bar{1}0\bar{1})_{C2}$ was established between the $C_2$ and $C_3$ domains. Thus, an 11.6° wedge disclination formed at the triple junction. In contrast, two wedge disclinations of equal magnitude but opposite directions were located at two triple

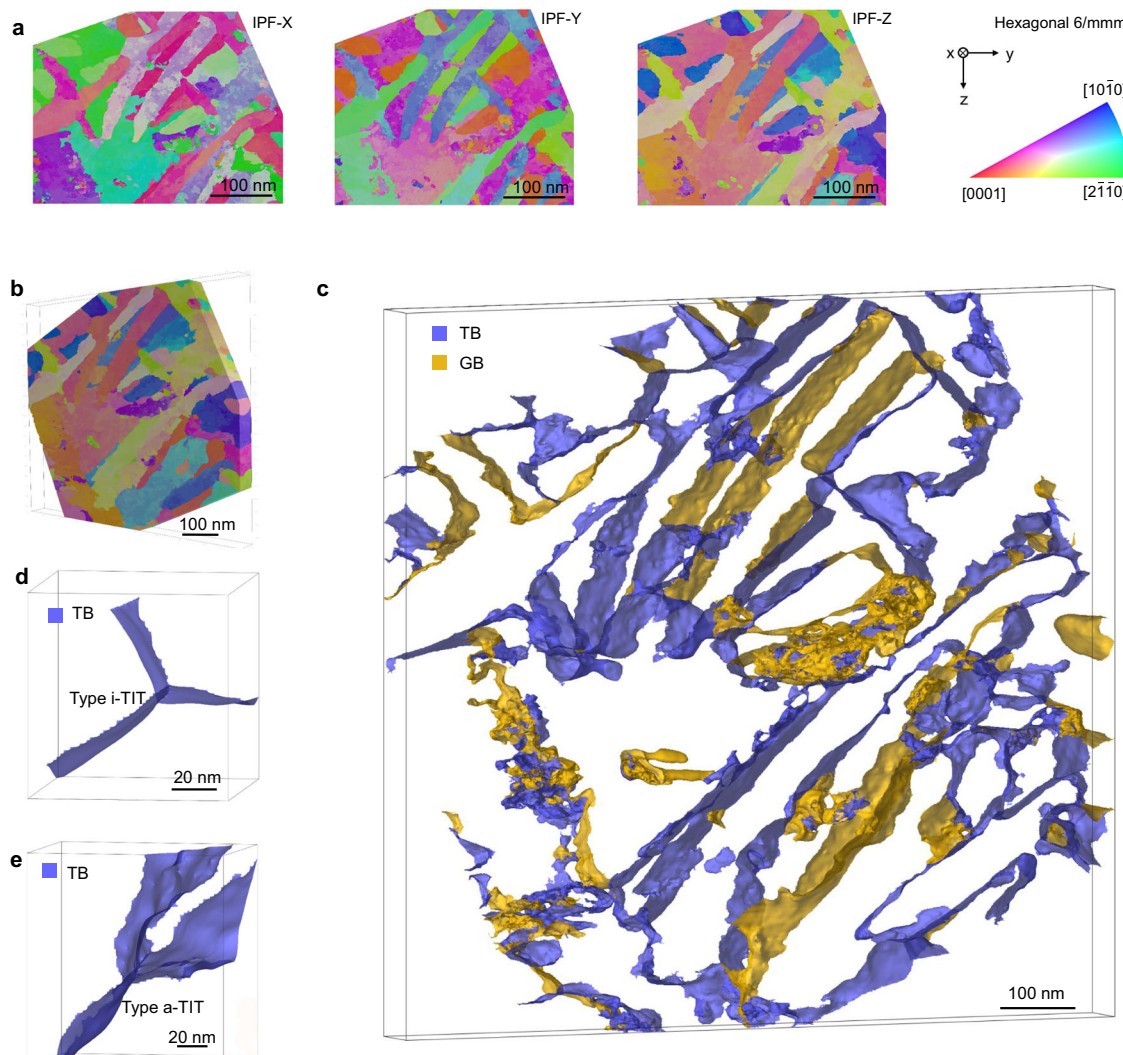

**Fig. 3 | 3D triple-interlocked-nanotwinned spatial structure. a** Plan-view of reconstructed 3D orientation map of the TIT structure. The coloring of three images refer to the crystal direction parallel to the X, Y, and Z directions of sample coordinate system, respectively. **b** Reconstructed 3D orientation map of the TIT structure. The coloring refers to the crystal direction parallel to the Z direction of sample coordinate system. **c** 3D reconstructed TIT structure inside Mg–Li sample. Blue and yellow parts represent twin boundaries and grain boundaries, respectively. Typical 3D reconstructed i-TIT structure and a-TIT structure are showed in (**d**) and (**e**), respectively.

junctions that were connected by the $C_2$-$C_3$ twin boundary. Considering another twin reaction configuration for $C_1$-$C_2$-$C_3$, two twin planes $(1\bar{1}0\bar{1})_{C1}$ and $(1\bar{1}01)_{C1}$ intersected at an acute angle, and a $(\bar{1}10\bar{3})_{C2}$ twin plane and a 3.7° disclination dipole formed at the intersection of $C_2$ and $C_3$.

Moreover, as evidenced by an HRTEM image (Fig. 2g), many $I_1$-type stacking faults (SFs) were detected in the matrix, which were formed by removing an A plane above a B plane and then shearing the remaining planes above the B plane by $1/6[20\bar{2}3]$, thereby resulting in a ...ABABAB CBCBCB... structure[19]. In addition, both $I_2$-type SFs caused by slipping of 1/3 of the $[10\bar{1}0]$ Schockley partial dislocations and an orientation deviation of 4.6° along the (0001) direction were confirmed. These deformed structures accounted for the homogeneous strain distribution during the construction of the TIT structure, as demonstrated by geometric phase analysis (GPA) (Supplementary Fig. 5). Additionally, we analysed randomly distributed interface structures (Supplementary Fig. 6) and observed that the TIT structures were uniformly distributed in the HPHT-800 Mg-9Li alloy. The interface structures were composed mainly of original GBs, TBs and basal-prismatic (BP) interfaces, and the average interface fraction of TBs was approximately 65.7% (Fig. 2h).

In addition, to further elucidate the three-dimensional (3D) configuration of TIT, the sample was characterized by 3D orientation mapping via TEM (3D-OMiTEM)[20]. Three-dimensional tomography was performed (Fig. 3a, b), and the HCP/BCC phase matrix was removed to clearly show the interface structure (see Supplementary Movie 1). A typical view (Fig. 3c) demonstrates that most of the TIT structures were net-like and continuous, analogous to GBs, and were randomly distributed throughout the alloy. In detail, a representative ellipsoid-shaped characteristic of the TIT structure is first detected (Fig. 3d, e), in which the fractions of i-TIT and a-TIT structures are approximately 42.85 and 57.15%, respectively (Supplementary Fig. 6). Moreover, the spatial width of twins (the maximum narrow value of spatial ellipsoid) is around 7 nm (Supplementary Fig. 7). Note that some isolated fragments were scattered in the matrix, suggesting that non-HCP phase structures (strain areas) were distributed along twin boundaries.

## TIT structure formation mechanisms

To probe the phase transformation process during the HPHT process, in-situ synchrotron X-ray diffraction (SXRD) tests (Fig. 4a) were performed on Mg-9Li alloys under a pressure of 6 GPa at

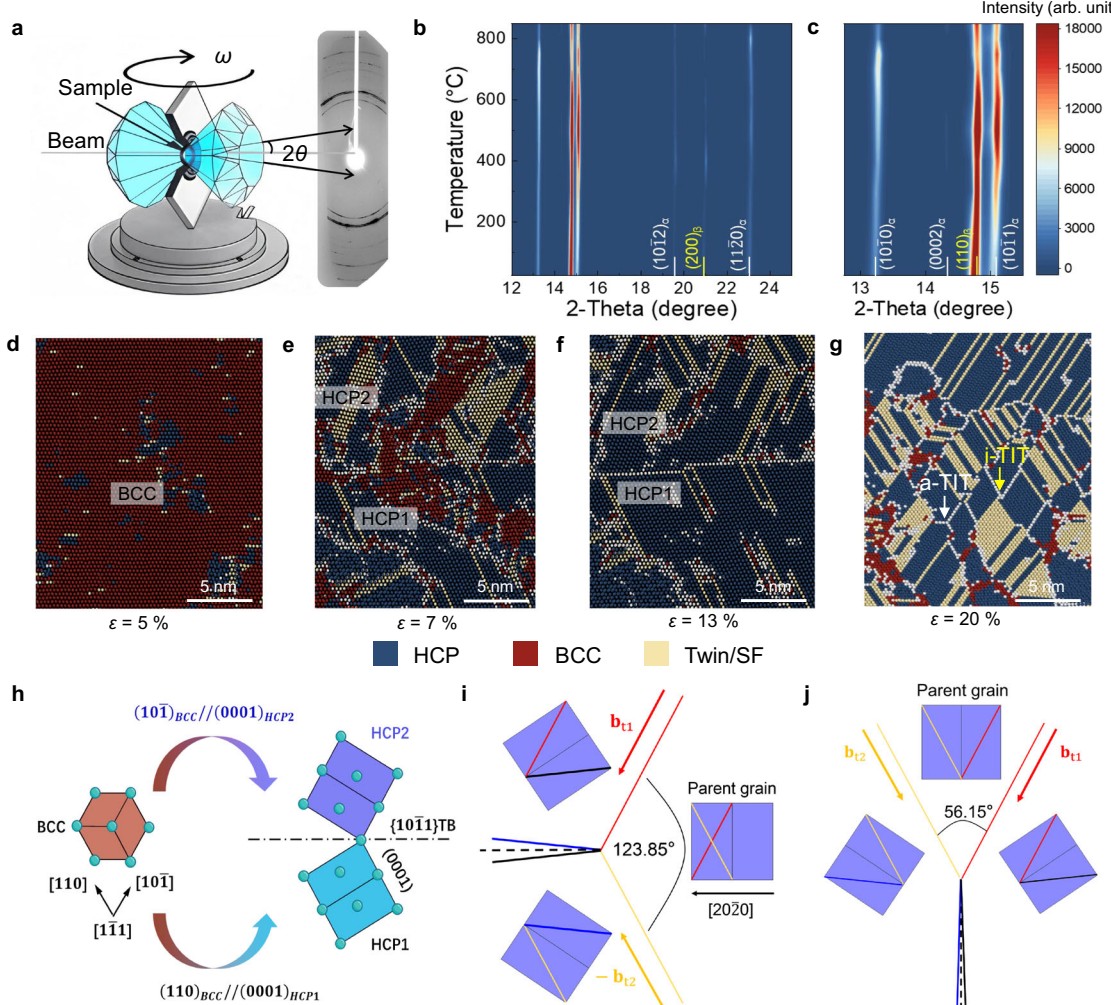

**Fig. 4 | Formation mechanism. a** Schematic diagram of the in situ SXRD experimental setup. **b** Temperature-dependent two-dimensional SXRD patterns. **c** Expanded patterns that show details in the range of 12.75°–15.5°. The coloured bars indicate the intensity. **d** Deformed Mg-9Li supercell. The blue region represents α-Mg with an HCP structure, the red region represents β-Li with a BCC structure, and the yellow region represents α-Mg that contains a TB/SF structure. **e** Two HCP phases with different orientations in the BCC region owing to Li diffusion. Tensile deformation was performed along the x-axis, and complementary compressive deformation was performed along the y- and z-axes at 0 K. **f, g** Twin variants that gradually nucleated and grew as the strain increased. ε represents the engineering strain. **h** Schematic illustration of the orientation relation of the BCC–HCP phase transformation. **i, j** Possible formation mechanisms of i-TIT (**f**) and a-TIT (**g**) from a crystallographic perspective. The MD data are provided at https://doi.org/10.24435/materialscloud:rf-56. Source data for (**b, c**) are provided as a Source Data file.

various temperatures. Owing to the limitation on the diamond tip oxidation temperature, the measured temperature range is below 850 °C. The results (Fig. 4b, c) show that the HPHTed Mg-9Li alloy retained two phases (HCP-α phase and BCC-β phase) throughout the entire HPHT process. Nevertheless, the localization of these phases changed markedly with increasing temperature, as verified by the variations in the $(10\bar{1}1)_\alpha$ and $(110)_\beta$ peak intensities. Specifically, the localization of the HCP-α phase gradually increased as the temperature increased, whereas the localization of the BCC-β phase decreased accordingly. A typical BCC-to-HCP solid-state martensite phase transformation occurred during the HPHT process, which was consistent with the theoretical prediction for sheared pure iron[21]. Additionally, this process of the BCC-to-HCP phase transformation was confirmed by quasi-in-situ XRD over a wider temperature range (Supplementary Fig. 8). A typical "V-shaped" trend was confirmed, in which the volume fraction of the BCC-Li phase first changed from 55% in the as-cast sample to 10.2% in the HPHT-800 sample with increasing temperature from room temperature to 800 °C. Afterwards, the volume fraction of the BCC-Li phase increased with increasing temperature, as revealed

by Rietveld refinement. Due to the difference in thermal means, size dimension, and pressure modes between SXRD and by quasi-in-situ XRD, the critical temperature value of phase peak concentration is slightly different, whereas the trend is the same.

To explore the twin formation process during phase transformation, we performed molecular dynamics (MD) simulations to investigate the microstructural evolution. The random solid-solution Mg-9Li alloy was composed of a BCC phase-dominated region and an HCP phase-dominated region. Numerous nanoscale BCC particles were homogenously embedded in the HCP phase-dominated region after structural relaxation. As the strain increased (Fig. 4d–g, Supplementary Fig. 9, and Supplementary Movie 2), the nanoscale BCC particles in the HCP phase-dominated region gradually disappeared. Afterwards, the phase transition initiated simultaneously from the top left and bottom right regions in the BCC phase to form two oriented HCP phases. The two newly formed HCP phases, HCP1 and HCP2, were misoriented by 57° around a common $<11\bar{2}0>_{HCP}$ axis, which corresponded to a $\{10\bar{1}1\}$ contraction twin relation in HCP materials. As deformation continued, the twins grew gradually, and the different twins interacted with each other to form TIT structures, as evidenced

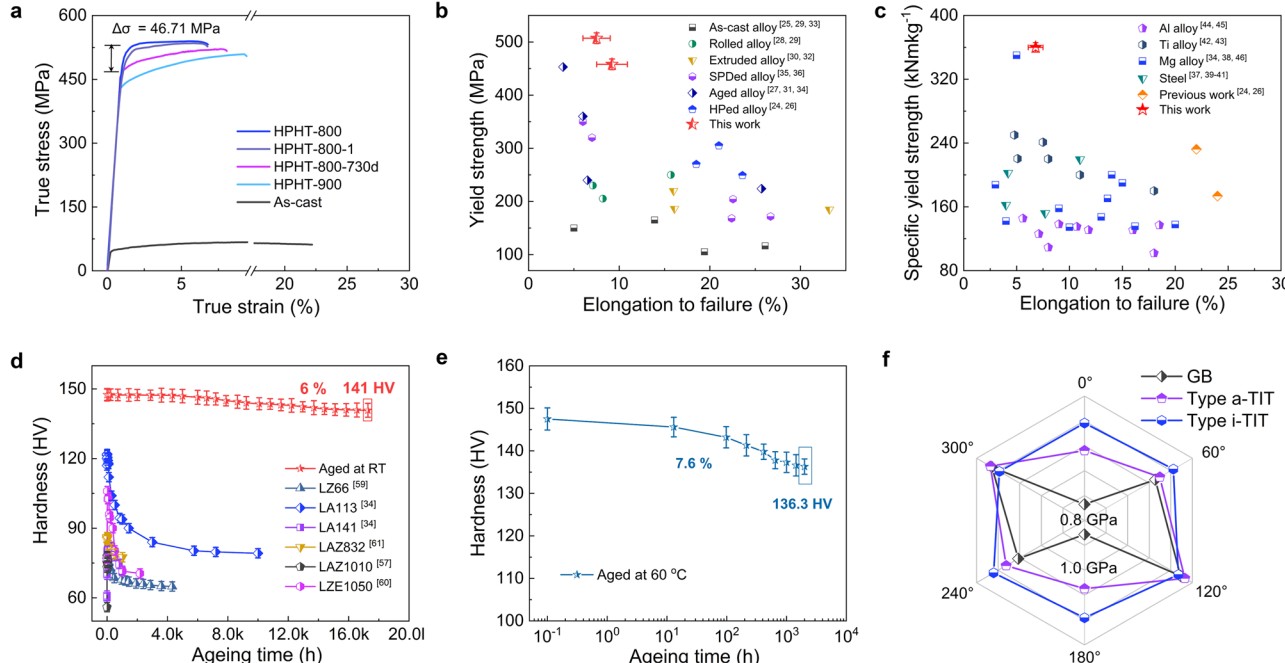

**Fig. 5 | Mechanical properties. a** Tensile true stress–strain curves of Mg-9Li alloys in various states at a nominal strain rate of $10^{-3}\,s^{-1}$. **b** Comparison of the tensile properties of the HPHT-800 Mg-9Li alloy and other existing Mg–Li-based materials. The cited data are listed in Supplementary Table 3. **c** Comparison of the specific yield strength and elongation of the HPHT-800 Mg-9Li alloy with those of other high-strength Ti alloys, Al alloys, steel and Mg alloys. The cited data are listed in Supplementary Table 4. Error bars in (**b**, **c**) represent the standard deviation derived from five independent experiments. **d** Unprecedented stability of the HPHT-800 Mg-9Li alloy in comparison with that of other reported Mg−Li alloys at ambient temperature. **e** The hardness curve of the HPHT-800 Mg-9Li alloy aged at 60 °C. Error bars in (**d**, **e**) represent the standard deviation of hardness derived from nine independent indentation points. **f** Comparison of the mechanical anisotropy among i-TIT, a-TIT, and GBs. The six corners of the radar chart correspond to 0°–360°, respectively. The center of the radar chart represents 0.8 GPa, and the interval between successive concentric hexagons is 0.1 GPa. The data are originated from Supplementary Fig. 23a–c and listed in Supplementary Table 5. Source data are provided as a Source Data file.

by the simulated 3D framework (Supplementary Fig. 10), which is consistent with the above 3D structure of TIT.

To further interpret the formation process of TIT structures, we elucidated these underlying processes on the basis of the variations in the crystallographic orientation relationship. Specifically, the basal planes of HCP2 and HCP1 originated from $(10\bar{1})_{BCC}$ and $(110)_{BCC}$ planes in the BCC phase that share a common $[1\bar{1}1]_{BCC}$ axis, as illustrated in Fig. 4h. Hence, the phase transition followed the Burgers mechanism: $\{110\}_{BCC} 0001\} \parallel_{HCP}$ and $[1\bar{1}1]_{BCC} 11\bar{2}0] \parallel_{HCP}$. Therefore, the orientation relationship in each HCP variant can be summarized as follows:

BCC-to-HCP1: $(110)_{BCC} 0001) \parallel_{HCP1}$, $[1\bar{1}1]_{BCC} 11\bar{2}0] \parallel_{HCP1}$;
BCC-to-HCP2: $(10\bar{1})_{BCC} 0001) \parallel_{HCP2}$, $[1\bar{1}1]_{BCC} 11\bar{2}0] \parallel_{HCP2}$.

This finding indicates that the formation of $\{10\bar{1}1\}$ contraction twins in Mg−Li alloys is closely associated with the BCC-to-HCP phase transformation, which is similar to the martensite process in titanium and iron[22,23]. Meanwhile, the crystallographic orientation relationship similar to that of martensitic transformation has been confirmed through MD simulations (Supplementary Fig. 11). However, differing from common martensite process in titanium and iron driven by shear at low temperatures or under stress, this martensitic-like transformation is triggered by the diffusion and redistribution of Li atoms during the HPHT process, as further confirmed by MD, TEM and atom probe tomography (APT) (Supplementary Figs. 12–16 and Supplementary Note 2). Additionally, TBs generally exhibit a specific orientation with respect to parent grains as well as other contraction twin variants (Fig. 4i, j and Supplementary Fig. 17). Both i-TITs and a-TITs can be explained by the interaction of two conjugated contraction twins in combination with their parent grains (Supplementary Note 3). For the i-TIT, two conjugated contraction twins impinge on each other and constitute a unique microstructure together with residual parent grains. Their twinning shear directions are different; i.e., one belongs to $\langle 10\bar{1}2 \rangle$, but the other belongs to $\langle 10\bar{1}2 \rangle$. The reaction leads to $\langle 20\bar{2}0 \rangle$ along the basal plane of the parent grain and near the $\{10\bar{1}1\}$ lattice plane of the active twins, as outlined in the $\{10\bar{1}1\}$ pole figure (Supplementary Fig. 18). In contrast, the a-TIT is composed of one parent grain and two conjugated contraction twins. Thus, the angle between two TBs that adjoin the parent grain is 56.15°, which is a supplementary angle with respect to the i-TIT type. The sum of these two twin variants is $\langle 000\bar{4} \rangle$ along the c-axis of the parent grain, which is approximately parallel to the $\{10\bar{1}3\}$ lattice plane of the active twins, as shown in the $\{10\bar{1}3\}$ pole figure.

## TIT deformation behaviours

The hardness of the pre-HPed HPHT-treated Mg-9Li alloy remained relatively stable and was close to the as-cast value (Supplementary Fig. 19). A volcano-like trend was detected between the hardness and temperature during the second step of treatment (6 GPa and 200–1000 °C). The peak hardness of the HPHT-800 alloy was 147.5 HV. The variation in the hardness (from 15 min to 2 h) indicated that the strengthening structures were very stable once they were formed. The tensile yield strength of the HPHT-800 sample was 508.6 MPa, which was approximately eleven times greater than that of the as-cast sample without TIT structures (Fig. 5a). Moreover, the compression yield strength is up to 534 MPa, which is slightly higher than tension yield strength (Supplementary Fig. 19). Compared with the as-cast sample, the Young's modulus is marginally improved (Supplementary Fig. 20). As plotted in Fig. 5b, the tensile yield strength of the HPHT-800 sample was not only far greater than those of Mg–Li binary alloys (∼200 MPa)[24–26] but also greater than those of Mg-Li-X-based alloys (100 ∼ 260 MPa)[27–34] and even other Mg−Li samples that were prepared

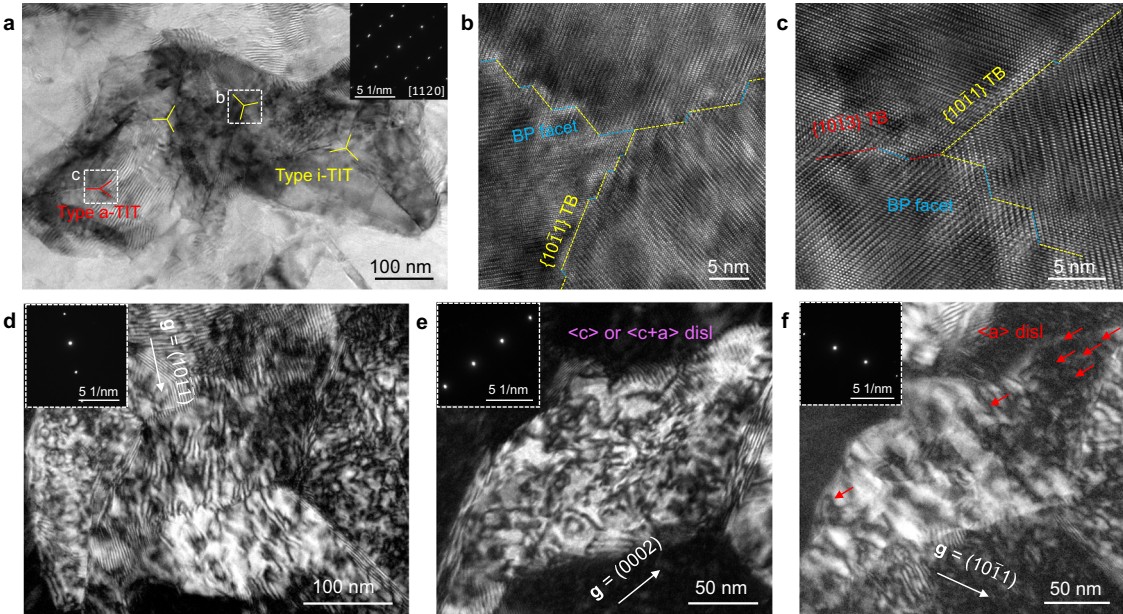

**Fig. 6 | Deformed characteristics. a** The morphology of the HPHT-800 Mg-9Li alloy after tensile fracture viewed along [11$\bar{2}$0] direction. The corresponding SAED pattern is inserted in top right corner. **b, c** HRTEM image of the deformed i-TIT and a-TIT structure, respectively. **d** Dual-beam dark-field image when $g = (10\bar{1}\bar{1})$. The corresponding SAED pattern is inserted in the upper left corner. A large number of dislocations have been pinned by TIT structures. **e, f** Dual-beam dark-field image when $g = (0002)$ and $g = (10\bar{1}1)$, respectively. The corresponding SAED patterns are inserted in the upper left corners. The results show that a large number of <c> and <c + a> dislocations have been activated during the deformation process.

via severe plastic deformation[35,36] (Supplementary Table 3). More importantly, owing to the high Li content, the highest specific yield strength of 350 kN m/kg was achieved, which was greater than those of most steels, Mg or Al alloys and even Ti alloys (Fig. 5c and Supplementary Table 4)[24,26,34,37–46].

To assess the stability of its mechanical properties, a typical natural ageing test was performed on the HPHT-800 sample. As summarized in Fig. 5d, the hardness of all reported Mg–Li-based alloys generally decreased by more than 20% after 500 h[31,47–50]. Conversely, the HPHT-800 sample displayed unprecedented ageing resistance, with a reduction in hardness of only 6% even after ageing for 730 days (approximately two years) at ambient temperature. A high yield strength (443 MPa; Fig. 5a) was attained. Moreover, even aged at 60 °C for 2000 h, a high hardness value (~136 HV) is detected (Fig. 5e). TEM characterization results demonstrate that the TIT strengthening structure possesses excellent structural stability under both long-term ambient and slightly elevated temperature conditions (Supplementary Fig. 21 and Supplementary Note 4). Therefore, compared to the traditional precipitation strengthening, TIT structure strengthening strategy offers a completely new design concept and mechanism for achieving Mg–Li-based alloys with both high strength and excellent long-term stability.

To explore the deformation process of perfect TIT structures, MD simulations were employed to explain the deformation processes of the i-TIT, a-TIT and GBs. As evidenced by the results (Supplementary Fig. 22 and Supplementary Movie 3), the GBs first become "zig-zag"-shaped during early deformation. The dislocations cause the GBs to move towards the interior of the matrix, thereby resulting in matrix deformation. Conversely, the initial strain cores lie in the matrix for TIT structures. The dislocations are produced primarily in the matrix, and their motion is constrained by TBs. Second, the GBs become ambiguous with increasing strain, and various dislocations directly cut through the GBs. In contrast, the typical triplet characteristic was observed in TIT structures, even when the strain was greater than 12%, which suggests that the TIT structures offer high mechanical stability. Eventually, several high-atom layer steps (ALSs) were detected in the

TIT structures, wherein these ALSs can prohibit dislocation motion around them. Additionally, to compare the strengthening anisotropy among the i-TIT, a-TIT and GBs, strain was evaluated *via* stress loading tests along various orientations (Fig. 5f, Supplementary Note 5, Supplementary Fig. 23, and Supplementary Table 5). The i-TIT structure exhibited the weakest strain anisotropy, in which the yield strength remained approximately 1.2 ± 0.1 GPa in all the loading directions. A similar trajectory was also observed in the a-TIT structure with a slightly lower yield strength. In turn, a typical dumbbell-shaped outline was detected at the GBs. These results demonstrate that the TIT structure can weaken the strengthening of the anisotropy.

Experimentally, as verified by in situ nanopillar compression tests (Supplementary Movie 4 and Supplementary Fig. 24), the HPHT-800 alloy with TIT structures exhibited a similar and extremely high stress of ~1.1 GPa, depending on two different random orientations, which was far greater than that of the as-cast sample with GBs (~400 MPa), suggesting that strengthening anisotropy was effectively eliminated. According to the Hall–Petch relationship (Supplementary Note 6), the yield stress ($\sigma_{GB}$) of the HPHT-800 alloy is approximately 50.6 MPa, which is similar to the yield strength of the as-cast alloy. In contrast, the contribution of TIT strengthening is approximately 451 MPa, as determined by nonuniform partial dislocation extension[51] and confined layer slip models (Supplementary Note 7)[52]. The strengthening mechanism of TIT structures is similar but far superior to that of traditional GBs.

Finally, the HPHT-800 sample significantly improves the yield strength without apparent sacrificing its ductility. The possible reasons lie in three aspects. Firstly, as evidenced by TEM observations, the formation of numerous high-stepped BP facets in TBs (Fig. 6a−c) can effectively alleviate stress concentration[14,28]. Secondly, TBs serve as effective barriers that can significantly facilitate dislocation accommodation (approximately $1.171 \times 10^{14}$ m$^{-2}$ in terms of Williamson-Hall analysis, Supplementary Fig. 25 and Fig. 6d). Thirdly, the high-density TIT structures contains HCP matrices with multiple orientations. This feature can provide favorable crystallographic orientation conditions for the activation of non-basal slip during deformation process. TEM

observations confirm the presence of a high density of non-basal <c + a> dislocations in the deformed TIT structure, whose density even exceeds that of basal <a> dislocations (Fig. 6e, f). Consequently, the barrier effect of TBs and the activation of non-basal slip are the primary mechanisms responsible for the high work hardening rate, especially under compression condition (Supplementary Fig. 19). From the perspective of the fracture surface, the observation of dimple features within grain interiors further supports the plastic deformation mechanism discussed above (Supplementary Fig. 26).

In summary, we propose the TIT concept to overcome the intrinsic deficiencies of twin engineering for structural metals. The formation of TIT structures is associated with three main features: a twinning matrix, a BCC-to-HCP martensite-like phase transformation, and a suitable external stress/temperature field. Considering these three aspects, we first achieved this unique structure in a Mg-9Li alloy through a straightforward industrial route. The results demonstrate that the TIT concept is a good strategy for optimizing twin engineering strengthening. For example, the universality of this design concept has also been extended to the HPHT Mg–Sc alloy (Supplementary Fig. 27), in which similar triple-interlocked interfaces, which are formed by the BCC-to-HCP solid-state martensite-like phase transformation, effectively impede dislocation motion to realize high mechanical properties. The limitation on the dimensions of nanotwinned materials, which depends on growth twinning or deformation twinning, has been well overcome by phase transformation twinning. A large-sized nanotwinned Mg–Li sample (with a maximum diameter of more than 40 mm) can be prepared for large-scale applications through current industrial setups.

Attractively, the mechanical properties of this material, especially its yield strength and ageing resistance, are greatly enhanced. Both of these parameters for this material are far better than those that have been reported for Mg–Li alloys thus far. Concurrently, the highest specific strength value is achieved because of the low density of the material. This combination of properties reveals that such nanotwinned Mg–Li binary alloys could be applied to a wider range of applications, e.g., as load-bearing components or as substitutes for other expensive high-strength Mg alloys. Compared with other heavily alloyed Mg alloys with similar mechanical properties, nanotwinned Mg–Li alloys exhibit low toxicity and stable potential. This implies that nanotwinned Mg–Li alloys could be further applied as anodes for Mg/Li-based batteries[53] or nontoxic degradable cardiovascular stents[54]. All of these distinctive characteristics demonstrate that TIT Mg–Li alloys are not only scientifically interesting but also potential industrial products.

## Methods
### Materials and sample preparation
An alloy ingot with a nominal Mg-9Li (wt.%) composition was prepared by induction melting in a steel crucible under argon protection and then cast into a steel mould. The raw materials were 99.99 wt.% pure Mg and 99.99 wt.% pure Li, which were purchased from Aladdin Reagent (Shanghai) Co., Ltd., China. Actual chemical compositions (wt. %) of as-cast and HPHT-800 Mg-9Li alloy determined by inductively coupled plasma optical emission spectroscopy (ICP-OES) are listed in Supplementary Table 1. The cast alloy was machined into cylindrical samples with a diameter of 30 mm and a length of 15 mm for HPHT treatment conducted in a cubic-anvil high-volume press equipped with six rams. The device was manufactured by the Guilin Metallurgical Machinery Factory in China. The samples were wrapped in tantalum foil and subsequently inserted into a thermally stable BN capsule. The samples were heated in a graphite furnace, and cubic pyrophyllite was used as the pressure medium. Hydrostatic pressure was applied by pressing along three axes. Two-step HPHT treatment was subsequently performed. The samples were first pretreated for 30 min at 400 °C and 1 GPa. After pressure pretreatment, the samples

were further treated at 6 GPa for various durations in the temperature range of 200–1000 °C. After the two-step HPHT treatment, the samples were quenched to room temperature directly before the pressure was unloaded. The sample that was subjected to HPHT treatment at 6 GPa and 800 °C was denoted as HPHT-800. The as-cast and other HPHT-treated samples were used as references. In addition, to investigate the universality of the design concept, a Mg–Sc alloy was subjected to the same preparation technique and HPHT treatment process.

### Mechanical properties
The microhardness was measured by an FM-ARS-9000 Vickers hardness tester with a load and dwelling time of 100 g and 15 s, respectively. The average hardness was calculated on the basis of the dependence on a $6 \times 6$ matrix. The tensile tests were performed at room temperature via a WDW-50S MTS universal material testing machine at a normal strain rate of $1 \times 10^{-3}$ s$^{-1}$. Dog bone-shaped samples with a gauge length of 18 mm and a cross-sectional area of $3 \times 3$ mm were used for tensile testing. The average value from at least five parallel samples was determined. In addition, nanopillars with a diameter of 300 nm were prepared by annular milling for FEI Heilos 5 CX focused-ion-beam (FIB-SEM). To tailor the shape and minimize the tapering of the nanopillars, a low beam current (30 kV and 40 pA) was used for milling throughout the entire process. All the nanopillars were prepared in random grains with a diameter-to-height aspect ratio of approximately 1:3. Before the samples were loaded into the microscope for compression, they were polished by Ar ion milling (NanoMill, Model 1040, Fischione) to remove surface damage. In situ nanopillar compression experiments were conducted with an FEI Titan ETEM G2 at 300 kV. The elastic modulus of as-cast and HPHT-800 sample was evaluated using the nanoindentation tests using the Hysitron TI-980 equipment. The applied indentation depth was 1000 nm, the loading rate was 100 nm/s and the hold time was 10 s.

### Structural characterization
The microstructural investigations were performed using optical microscopy (OM) and scanning electron microscope (SEM), where the samples were prepared by a procedure involving grinding up to 2000 SiC paper, followed by mechanical polishing with 9, 3, and 1 μm water-free diamond suspensions and finally polished using 0.05 μm colloidal silica. The final step included chemical polishing in a fresh solution containing a mixture of 100 mL alcohol and 5 mL nitric acid. For electron back scattering diffraction (EBSD) examinations, both the as-cast and HPHT-800 Mg-9Li samples were ground and electro-polished in the electrolyte (80% C₂H₅OH + 20% HClO₄) at 20 V for ~30 s and at −30 °C. EBSD was conducted over cross-sections (TD plane) of the samples using the JEOL JSM-7800 instrument equipped with an HKL-EBSD system (Oxford Instruments, UK) with step size of 0.09 μm. An Oxford Symmetry S2 detector was utilized for the EBSD system. The EBSD results were analyzed by the Channel 5 software (version 5.0.9.0). The mis-orientation angles between the adjacent grains are used to identify the low angular grain boundary ($2° \leq \theta \leq 15°$) and high angular grain boundary ($\theta \geq 15°$), as indicated by green and black lines, respectively. The average grain sizes were estimated from the inverse pole figure maps by using the major axis of fitted ellipse (with the software of Channel 5). The samples were characterized by TEM to reveal the microstructural evolution at the nanoscale during the HPHT treatment. Samples for TEM observation were extracted from the as-cast, HPHT-500 and HPHT-800 by using FIB-SEM (FEI Heilos 5 CX) at a voltage of 30 kV and an ion dose of $2 \times 10^9$ ions/μm$^2$ and then mounted on a dedicated half copper grid and thinned to 500 nm. The samples were further milled to a thickness of less than 100 nm and polished by Ar ion milling to remove surface damage. Finally, an FEI Talos-F200X transmission electron microscope that was operated at 200 kV was used to collect BF, HRTEM, and HAADF-STEM images. The atomic-

scale STEM images were obtained from a aberration-corrected FEI Themis Z equipment operated at 300 kV.

## Three-dimensional orientation mapping in TEM (3D-OMiTEM) characterization

Specimens for 3D-OMiTEM were prepared by a conventional FIB lift-out procedure. A tilt series from −68° to +62° in 1° steps was performed on a JEOL JEM-2100 (200 kV) instrument that was equipped with an EM-21010HTR single-tilt holder. At each tilt angle, automated dark-field (DF) conical scanning was executed with EM-DFOM module inside EM-TOOLS software (version 1.9.5.3 α, TVIPS). The incident beam was successively tilted onto the $\{10\bar{1}0\}$, $\{10\bar{1}1\}$, $\{10\bar{1}2\}$, $\{11\bar{2}0\}$, $\{10\bar{1}3\}$, $\{20\bar{2}0\}$, $\{20\bar{2}1\}$, $\{0004\}$, and $\{20\bar{2}2\}$ Mg diffraction rings; for every ring, the beam was azimuthally swept through 0–360° in 2° increments. DF images were recorded on a TVIPS F416 CCD camera with an exposure duration of 200 ms, which yielded more than 200,000 images for the dataset. Fiducial-based alignment of the tilt series was performed with an in-house MATLAB script. The DF images were binarized via machine learning-assisted segmentation[55], and diffraction vectors were extracted. Three-dimensional reconstruction followed the 3D-OMiTEM workflow, and a dictionary-based forwards indexing algorithm was used to assign voxel-scale orientations[55–58]. The grains were segmented with a 5° misorientation threshold, and the resulting 3D grain-boundary network was visualized with Avizo software (version 2022.1, Thermo Fisher Scientific).

## Atom probe tomography

Samples of the as-cast and HPHT-800 alloys were cut into dimensions of $10 \times 10 \times 2$ mm for APT sample preparation and subsequent analysis. Before FIB milling, the samples were mechanically polished using 9, 3, and 1 μm water-free diamond suspensions, followed by final polishing with 0.05 μm colloidal silica. They were then etched with a freshly solution containing a mixture of 100 mL alcohol and 5 mL nitric acid. APT tips were fabricated using the ring milling technique in FIB system. The APT analysis was performed on a Cameca LEAP-5000XR instrument, with parameters set to a pulse frequency of 120 kHz, pulse energy of 60 pJ, at a temperature of approximately 30 K, and in a vacuum of $\sim 2.5 \times 10^{-11}$ torr. Tomographic reconstruction and analysis were carried out using the IVAS software (version 3.8.2) package from CAMECA.

## In-situ synchrotron X-ray diffraction measurements

In-situ compression tests were performed by using a deformation-DIA apparatus coupled with synchrotron X-ray irradiation at beamline BL15U1 of the Shanghai Synchrotron Radiation Facility. For in situ SXRD experiments, samples were machined into sheets with a diameter of 10 mm and then ground with SiC paper to a thickness of 30 μm. Finally, several small pieces were cut from each sheet and placed into the sample chamber of a diamond anvil cell that was equipped with a resistance heating system and a circulating water system. The dimensions of the sample chamber were $\Phi 250 \times 100$ μm. The sample was assembled in a glove box. The energy and wavelength of the beamline were 20 keV and 0.6199 Å, respectively. $CeO_2$ was used to calibrate the sample-to-detector distance, and the detector was tilted. The collected two-dimensional diffraction patterns were integrated into one-dimensional profiles by using Dioptas software (version 0.6.1)[59]. The sample-to-detector distance and detector tilt angle were approximately 137.66 mm and 0.729°, respectively, according to the calibration data. Argon gas was injected into the chamber before the in situ experiments to prolong the life of the diamond and reduce heat leakage. During the experiments, the samples were treated at temperatures that ranged from 25 to 850 °C with a heating rate of 10 °C/min and a pressure of 6 GPa. Diffraction patterns were recorded over the 2θ range of 12°–25° with an exposure duration of 60 s. The results were analysed with Dioptas software.

## Density functional theory calculations

To elucidate the effect of Li content, the energy of Mg–Li alloys was investigated by first-principles calculations with the Vienna ab initio simulation package[57,60]. All configurations with varying Li contents are investigated using a $2 \times 2 \times 1$ supercell. The electron-ion interaction was described by the projector augmented-wave method[55,61]. The generalized gradient approximation with the Perdew–Burke–Ernzerhof functional for the exchange-correlation interaction was adopted[58]. The energy cutoff of 650 eV, the energy convergence of $10^{-4}$ eV/cell, and the 1st order Methfessel–Paxton method with a smearing witdh of 0.2 eV were chosen for electronic self-consistency calculation. A force convergence criterion of 0.01 eV/Å was used for ionic relaxation. The $k$-point meshes were chosen to be inversely proportional to the real space length in that direction so the ratio of 20 k-points to the lattice parameter was approximately maintained.

## Molecular dynamics simulations

The twin strengthening mechanism and the formation of TIT interfaces were simulated by MD with the large-scale atomic/molecular massively parallel simulator package[62]. The second nearest-neighbour modified embedded-atom method (2 nm MEAM) potential was chosen to describe the interatomic interactions[63], as it has been successfully used to investigate the mechanical behaviours of Mg-9Li alloys[34,64]. The simulation results were visualized and analysed via the open visualization tool)[65]. The common neighbour analysis algorithm was used to identify the BCC, HCP, and FCC structures and SFs in the HCP structure[66]. We used Atomsk to construct models with i-TIT, a-TIT and GBs[67]. Initially, a supercell that featured a 20 nm thick vacuum layer was employed to eliminate the internal stress that arose during artificial modelling. The dimensions of the supercell were $0.6 \times 80.0 \times 80.0$ nm³ ($x \times y \times z$), and the model dimensions were $0.6 \times 60.0 \times 60.0$ nm³. After relaxation for 60 ps with an isothermal–isobaric ensemble, the model was trimmed down to dimensions of $0.6 \times 45.0 \times 45.0$ nm³. To avoid the effects of periodic boundary conditions, the atoms within the outermost 0.5 nm layer along the $y$- and $z$-axes were anchored as fixed layers. Next, shear deformation was performed along the $y$-axis at a shear strain rate of $1.0 \times 10^{-3}$ ps⁻¹ with a maximum strain of 0.2. A canonical ensemble was employed to maintain a constant temperature. Random solid-solution models of Mg-9Li were also constructed via Atomsk code. For the Mg-9Li system, we employed a large-scale supercell with dimensions of $7.9 \times 44.1 \times 46.1$ nm³ along the $x$-, $y$-, and $z$-axes. To simulate the HPHT process, tensile deformation was performed along the x-axis, and complementary compressive deformation was performed along the y- and z-axes. The tensile strain rate was $1.0 \times 10^{-3}$ ps⁻¹, with a maximum strain of 0.6. An isothermal–isobaric ensemble was employed to maintain a constant temperature. Furthermore, a BCC supercell similar to HPHPT process with dimensions of $10.0 \times 10.0 \times 10.2$ nm³ ($x \times y \times z$) was constructed using to investigate the phase transformation from BCC to HCP in Mg–Li alloys, where the $x$-, $y$-, and $z$-axes were oriented along the $[1\bar{1}1]$, $[110]$, and $[\bar{1}12]$ directions, respectively. Following structural relaxation, uniaxial tensile deformation was applied along the loading direction at a constant strain rate of $1.0 \times 10^{-3}$ ps⁻¹ with the maximum strain reaching 0.2. The isothermal–isobaric ensemble was employed to maintain a constant temperature.

## Reporting summary

Further information on research design is available in the Nature Portfolio Reporting Summary linked to this article.

# Data availability

The data that support the findings of this study are available within this article and its Supplementary Information. The MD and DFT data are available at https://doi.org/10.24435/materialscloud:rf-56. Source data are provided with this paper. Additional data are available from

corresponding authors upon request. Source data are provided with this paper.

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

## Acknowledgements

Q.P. discloses support for the research of this work from the Fundamental and Interdisciplinary Disciplines Breakthrough Plan of the Ministry of Education of China (grant number JYB2025XDXM215), the National Natural Science Foundation of China (grant number 52331003 and 52527801), the Natural Science Foundation of Hebei Province (grant number C2022203003) and the Ministry of Education Yangtze River Scholar Professor Program (grant number T2020124). Y.T. discloses support for publication of this work from the National Natural Science Foundation of China (grant number 5288102). G.Z. discloses support for the research of this work from the National Natural Science Foundation of China (grant number 52471050). We acknowledge the support from Shanghai Synchrotron Radiation Facility (SSRF) for use of the beamline. In addition, we also thanks Prof. Yunchang Xin (Nanjing University of Technology) and Prof. Yuan Wu (University of Science and Technology Beijing) for EBSD and APT tests, respectively.

## Author contributions

Q.P. and Y.T. conceived the project. L.Z., J.W., A.N., Y.S., T.H., W.H., W.C., and W.Z. carried out the materials syntheses and structural characterizations. G.Z., J.W. and L.Z. conducted the mechanical measurements. K.T., Y.G., and B.Y., carried out computational investigation and provided theoretical analysis. L.W., and L.Z. conducted in situ synchrotron X-ray diffraction tests. B.Y. analyzed twin orientation. W.C., T.H and W.Z carried out the 3D image reconstruction. Q.P. and J.W. wrote the manuscript with assistance from co-authors. L.W. and Y.T. reviewed the paper. Q.P. was responsible for the overall direction of the project. All the other authors participated in preparing the manuscript and contributed to the discussion. Q.P., L.Z., and J.W. contributed equally to this work.

## Competing interests

Q. Peng, G. Zou, J. Wang, L. Zhou and Y. Sun are inventors on granted patents related to this work (CN202511282665.0, JP52502912056, US 19/444124 and NL4000810). The other authors declare no competing interests.
