## [Transparent Peer Review file · Nature Communications]

Triple-interlocked-nanotwinned bulk magnesium alloys with exceptional strength and ageing resistance

Corresponding Author: Professor Qiuming Peng

Version 0:

Reviewer comments:

Reviewer #1

(Remarks to the Author)

This manuscript reports a triple-interlocked nanotwinned (TIT) architecture formed in a bulk Mg-Li binary alloy by high-pressure high-temperature processing. The experimental workload is substantial, and the authors present a broad set of structural and mechanical characterizations. The high strength and apparent aging stability are potentially valuable for developing high-specific-strength magnesium alloys. However, the key issues concerning phase constitution, transformation mechanism, Li distribution, strengthening behavior, deformation response, and stability verification are not sufficient, and the process complexity of high pressure and high temperature treatment and small sample size of Mg-10Li binary alloy affect its application prospect.

(1) The Mg-10Li composition is located close to the single β -Li phase region (>10.3 wt.% Li) according to the phase diagram. Yet, the as-cast microstructure in the manuscript contains a considerable amount of α -Mg. This discrepancy raises concerns about composition accuracy and homogeneity. A reliable compositional verification and explanation for the excessive α -Mg is needed.

(2) The distinction between dendritic structure in the as-cast condition and equiaxed grains after HPHT is not clearly demonstrated. Methods used to identify grain boundaries versus dendrite boundaries, and how grain size and its distribution were quantified, need to be explicitly described. Larger-area EBSD mapping is strongly recommended.

(3) The rationale for selecting Mg-10Li should be better justified based on thermodynamics and the phase diagram. Since Mg-10Li contains minor α -Mg, some of the current mechanistic explanations may not hold and require reconsideration.

(4) The manuscript attributes the BCC \rightarrow HCP microstructure evolution to a martensitic transformation, yet the specific transformation conditions and how this process compares with known martensitic mechanisms in Mg-Li alloys are insufficiently explained. A clearer mechanistic description is needed.

(5) The crystallographic orientation relationships for BCC \rightarrow HCP transformation are described but lack direct experimental proof. High-resolution diffraction or HRTEM/FFT indexing evidence should be provided to confirm such relationships.

(6) Several twin-boundary HRTEM images are insufficiently resolved to identify the exact atomic configuration. Improved atomic-resolution STEM characterizations are needed to verify the proposed interface structures.

(7) Since multiple crystal structures (BCC, FCC, HCP) are mentioned, quantitative elemental distribution data are essential, particularly regarding Li partitioning. The absence of APT/EELS or reliable STEM-EDS analysis is a critical gap that must be addressed.

(8) If both primary α -Mg and transformation-induced HCP phases are present, their compositional and crystallographic differences must be clarified. Distinguishing these two is necessary to validate the transformation interpretation.

(9) The claim that twin structures inhibit crack initiation and propagation is unsupported because no fracture behavior characterization is provided. Fractography and crack-path analysis are required.

(10) The manuscript provides extensive characterization and modeling to explain the strength improvement, but offers very limited discussion regarding tensile ductility. In current Mg-Li alloys, achieving ultra-high strength is typically accompanied by a drastic loss of elongation, and mechanical testing is usually limited to compression conditions. In contrast, the authors report relatively good tensile ductility while maintaining high strength, which is highly unusual for this alloy system. A more detailed mechanistic explanation of how ductility is preserved-considering deformation modes, strain hardening behavior, slip/twin activity, and potential detwinning-is necessary to substantiate this result.

(11) The claim of two-year aging resistance requires stronger evidence, including microstructural characterization after aging and evaluation at slightly elevated temperatures. The current analysis is insufficient to establish long-term stability.

(12) Considering the HPHT temperature approaching $\sim 800^{\circ}\text{C}$, possible changes in alloy composition, especially Li evaporation or burn loss, must be evaluated and reported.

- (13) The manuscript reports much higher compressive strength than tensile strength, yet Mg-based alloys are known for strong tension-compression asymmetry. Discussion on this asymmetry and the potential need for in situ tensile testing is required.
- (14) Dislocation behavior, including type and density, is not analyzed despite its importance in twin-induced strengthening. TEM-based identification and XRD-based density estimation should be included.
- (15) The lack of Li characterization throughout the paper is a major issue because Li plays a critical role in Mg-Li alloys. Proper detection and mapping of Li are necessary to support any structural or mechanistic claims.
- (16) The effect of transformation on density and elastic modulus should be considered, since these properties are crucial for Mg-Li alloys. Direct measurements before and after HPHT are needed.
- (17) The transformation-induced HCP structure may represent a metastable supersaturated solid solution given the alloy's phase diagram position. Thermodynamic reasoning supporting its long-term stability is needed.
- (18) The FCC-ordered structure mentioned is neither clearly identified nor supported by XRD evidence. Detailed crystallographic characterization and explanation of its formation are needed.
- (19) The claimed reduction of anisotropy requires experimental proof through mechanical testing in multiple orientations or directions. Current evidence is insufficient.
- (20) The aging resistance in this work should not be directly compared with precipitation-strengthened Mg-Li alloys, because the underlying aging mechanisms are fundamentally different. For binary Mg-10Li alloys, no aging softening typically occurs from a mechanistic perspective. However, the alloy studied in this manuscript exhibits a certain degree of softening (~6%), which contradicts the statement of excellent aging resistance. The validity of this performance claim therefore needs further justification.
- (21) The in situ synchrotron experiment conditions appear different from actual HPHT processing. The authors should explain how well the in situ observations represent the real processing conditions and microstructure evolution.

Reviewer #2

(Remarks to the Author)

The authors present the mechanisms leading to the creation of triple-interlocked-nanotwins (TIT) in Mg-Li alloys based on phase transformation, their interactions with dislocations during plastic deformation and their impact on the mechanical properties. The underlying mechanisms are unraveled using in-situ testing and molecular dynamics modelling. This work emphasizes the very high strength and the stability over time of the alloys. The mechanisms observed that lead to phase transformation twinning are not new, but this work provides new information about the creation of TIT network within MG-Li alloys. However, before publishing, I recommend addressing all the following comments concerning the content and some of the figures:

1. Fig 1.b: please zoom in on one or 2 grains to clarify you schematic
2. Please explain more carefully your initial microstructure as well as the HPHT one in terms of phase proportions, crystallographic texture, grain size, phase morphology ... The role of the Li-rich BCC phase seems to be key in the TIT processed but the localization of this minor phase within the matrix is not clear. Also how do the authors explain the presence of TIT within HCP regions that do not face phase transformation? Please clarify that in the manuscript. Without that, the importance of manufacturing a duplex phase alloy cannot be understood clearly.
3. Fig 2a: please increase its size or crop part of the image to enlarge the view to enable the visual identification of the TIT structures by the reader. Can you also provide the related diffraction pattern?
4. Can you provide more details about the twin thickness measurement? How many grains were considered? Are they statistically relevant of the whole microstructure?
5. Does the crystallographic orientation of the grains affect the TIT formation? on fig 2a, some grains seem to not contain TIT network. Can you also comment on the role of the misorientation between grains in that process?
6. What is the percentage of i-TIT and a-TIT in your material? Can it be controlled somehow? Does it affect the overall plastic mechanisms and mechanical properties of the alloy?
7. Lines 109 to 111: the statement about the density of the structure is not clear. Can you quantify the different densities measured and explain how they were measured?
8. Figure 2k: a GB proportion is mentioned while no GB is visible on the 12 analyzed structures. Can you clarify that?
9. Please clarify the role of stacking faults on the TIT formation. Why do MD simulations not emphasize the role of stacking faults?
10. You present hardness variation over 2h while comparing later the tensile properties of the alloys after 2 years of ageing. What is the corresponding hardness after 2 years of ageing? Also how do you explain the drop of hardness when a temperature higher than 800°C is used for the second step of HPHT? Please also explain why no difference in hardness is observed after the first step of HPHT?
11. Fig 4a: the stress is not supposed to decrease in a true-stress/true-strain graph. Can you please remove the data obtained after strain localization arises in the tensile specimens? Can you also detail how the yield stress was calculated?
12. The strain hardening observed on the specimen deformed after 2 years of ageing seems higher than the hardening observed on a "fresh" specimen. Can you comment? How are affected the TIT-dislocation interaction mechanisms?
13. Why do your MD simulations not predict the role of FCC structures during the deformation process? To what extent is the microstructure used to model the alloy comparable to the experimental microstructure observed? Can you explain the modelling strategy used to generate the initial microstructure?
14. Insufficient information is provided on nanopillar compression experiments. Do they contain GB? Does the sample preparation affect the TIT structures? What is the crystallographic orientation of the pillar?

Reviewer comments:

Reviewer #1

(Remarks to the Author)

The authors have revised the manuscript according to the comments of the reviewers, I have gone through the manuscript and am satisfied with the revision. I am pleased to recommend the paper for publication.

Reviewer #2

(Remarks to the Author)

The authors have clarified the different points and questions during the first review stage.

Reply to the report of referee 1

1. The Mg-10Li composition is located close to the single β -Li phase region (>10.3 wt.% Li) according to the phase diagram. Yet, the as-cast microstructure in the manuscript contains a considerable amount of α -Mg. This discrepancy raises concerns about composition accuracy and homogeneity. A reliable compositional verification and explanation for the excessive α -Mg is needed.

Response: Thanks for the very careful review. We acknowledge your concern regarding the difference between the optical morphology in Supplementary Fig. 1 and the reported Mg-10Li composition. This is an oversight in our work, and we provide the following clarification. First, Mg-10Li is the nominal composition, and we initially lack verification of the actual alloy composition. Second, to address this point, we have performed additional ICP-OES analysis on the as-cast alloy. The results indicate an actual Li content of 8.87 wt.%. Such a deviation relative to the nominal composition is likely attributable to the evaporation and burn-off of Li during the melting process. Consequently, all references to the alloy composition in the manuscript have been updated to Mg-9Li. The relevant descriptions have also been updated.

We have made the following modifications to the manuscript:

“Actual chemical compositions (wt. %) of as-cast and HPHT-800 Mg-9Li alloy determined by inductively coupled plasma mass spectrometry (ICP-OES) are listed in Supplementary Table 1.”

Supplementary Table 1. Actual chemical compositions (wt. %) of as-cast and HPHT-800 Mg-9Li alloys by ICP-OES.

Alloy (wt.%)	Mg	Li	Fe	Ni	Cu	Bal.
As-cast	91.11	8.87	0.005	0.001	0.002	0.012
HPHT-800	91.34	8.63	0.005	0.002	0.003	0.020

2. The distinction between dendritic structure in the as-cast condition and equiaxed grains after HPHT is not clearly demonstrated. Methods used to identify grain boundaries versus dendrite boundaries, and how grain size and its distribution were quantified, need to be explicitly described. Larger-area EBSD mapping is strongly recommended.

Response: Thank you for your constructive suggestion. We have supplemented the EBSD characterization for both the as-cast and the HPHTed alloys. All grain size statistics in the manuscript have been updated based on the analysis of EBSD data. The EBSD results for the as-cast Mg-9Li alloy and the HPHT-800 Mg-9Li alloy are provided in the updated Supplementary Fig. 1. The relevant EBSD experimental sections have also been updated. According to EBSD results, the dendrite structure and equiaxed grain morphology can be well distinguished. In addition, it is important to note that the relatively low indexing rate for HPHT-800 sample is primarily attributed to the high internal stresses induced by the high density of nanotwins.

We have made the following modifications to the manuscript:

“For electron back scattering diffraction (EBSD) examinations, both the as-cast and HPHT-800 Mg-9Li samples were ground and electro-polished in the electrolyte (80% C₂H₅OH+20% HClO₄) at 20 V for ~30 s and at -30 °C. EBSD was conducted over cross-sections (TD plane) of the samples using the JEOL JSM-7800 equipped with an HKL-EBSD system with step size of 0.09 μm. The EBSD results were analyzed by the Channel 5 software. The mis-orientation angles between the adjacent grains are used to identify the low angular grain boundary (LAGB, $2^\circ \leq \theta \leq 15^\circ$) and high angular grain boundary (HAGB, $\theta \geq 15^\circ$), as indicated by green and black lines, respectively. The average grain sizes were estimated from the inverse pole figure (IPF) maps by using the major axis of fitted ellipse (with the software of Channel 5).”

Supplementary Figure 1 | Microstructure of Mg-9Li samples. **a**, Surface morphology and dimension of HPHT samples. The diameter and height are 30 mm and 15 mm, respectively. **b-c**, IPF image and the distribution of β -Li phase (blue) of the as-cast sample. **d-e**, Optical structure, IPF image, and pole figures of the HPHT-800 sample. **f**, Grain size variation of the different HPHT Mg-9Li samples.

3. The rationale for selecting Mg-10Li should be better justified based on thermodynamics and the phase diagram. Since Mg-10Li contains minor α -Mg, some of the current mechanistic explanations may not hold and require reconsideration.

Response: Thank you for your valuable suggestion, which prompted us to clarify the rationale for the composition selection more rigorously. The core theoretical basis for selecting the Mg-9Li (wt.%) alloy stems from its composition precisely within the α -Mg and β -Li two phase region (approximately 5.7-10.3 wt.% Li) of the equilibrium phase diagram. This specific compositional range is a necessary condition for realizing the core mechanism of this work. We will clarify this from the following two aspects.

Firstly, the martensitic transformation is dominated by the diffusion of Li atoms. According to the APT results (Supplementary Fig. 15), the Li concentration in the primary α -Mg is very low (3.3 at. %). Therefore, during the HPHT process, the primary α -Mg is the main site for Li atom diffusion (Supplementary Fig. 13). Furthermore, in single-phase Mg-Li alloys (such as Mg-13Li), the absence of Li-poor regions makes it

difficult for Li redistribution through compositional fluctuations to generate multiple HCP variants, thereby hindering the formation of the TIT structure. Our previous study [Ref. R1] showed that the HPHT-processed Mg-13Li alloy formed only double {10-11} twins, whereas no TIT structure was observed. This confirms the decisive role of the duplex phase region composition in forming TIT structure. Theoretically, alloys with compositions within this two-phase region can form the TIT structure after HPHT processing.

Secondly, a sufficient volume fraction of the β -Li phase not only ensures adequate transformation driving force during HPHT processing *via* local composition fluctuations (Li diffusion) but also aligns with the objective of reducing alloy density by increasing Li content to enhance specific strength. Therefore, the nominal Mg-9Li composition represents a result of synergistic optimization between the thermodynamic mechanism and performance goals.

[R1] Peng Q, Sun Y, Wang J, et al. Structural characteristics of {10-11} contraction twin-twin interaction in magnesium [J]. *Acta Mater.* **192**, 60-66 (2020).

Supplementary Figure 15 | APT analysis of as-cast sample. a1-a3, Reconstructed APT volume showing the distribution of Mg and Li atoms. **b1-b2**, APT results showing the distribution of Li atoms defined by 10 at.% and 23 at.% Li iso-surfaces, respectively. **c**, Concentration profiles along the red arrow in **b1**. **d-e**, Concentration profiles corresponding to 10 at.% and 23 at.% Li iso-surfaces, respectively.

4. The manuscript attributes the BCC→HCP microstructure evolution to a martensitic transformation, yet the specific transformation conditions and how this process compares with known martensitic mechanisms in Mg-Li alloys are insufficiently explained. A clearer mechanistic description is needed.

Response: Thank you for your valuable comment. The BCC→HCP microstructure evolution observed in this work is an analogous martensitic transformation driven by compositional fluctuations.

The specific conditions and mechanisms are as follows:

Unlike conventional martensitic transformations driven directly by shear at low temperatures or under stress, the transformation in this study is triggered by the diffusion and redistribution of Li atoms during the HPHT process. This argument is supported by XRD refinement results (Supplementary Fig. 8). If the transformation was only driven by shear, the BCC β -Li phase would be expected to transform into an HCP-structured Li phase. However, no corresponding diffraction signals for an HCP-Li phase were detected in XRD patterns. Furthermore, according to supplementary first-principles calculations (Supplementary Fig. 12b), the free energies of the BCC and HCP phases intersect at a Li concentration of approximately 23 at.%. This implies that when the local Li concentration exceeds this critical value, the BCC phase is more stable. Conversely, when the concentration falls below this value, the system spontaneously transforms from BCC to HCP via a martensitic transformation. Meanwhile, we have quantified the Li concentration changes within two phases during the MD simulation process (Supplementary Fig. 12a). Moreover, the reconstructed APT results indicated that the average Li concentrations are 3.3 at.% for the primary

α -Mg phase and 20.2 at.% for the β -Li phase in the as-cast sample (Supplementary Fig. 15). In contrast, the analysis of distinct regions reveals an average Li concentration of 7.3 at.% for the Li-poor HCP variants and 14.5 at.% for Li-rich HCP variants in the HPHT-800 sample (Supplementary Fig. 16). Both experimental and simulation evidence further supports that the martensitic-like transformation is primarily driven by Li compositional fluctuations.

During the HPHT process, high temperature provides a significant driving force for Li diffusion, while high pressure likely further promotes Li redistribution by affecting the chemical potential gradient. This leads to compositional fluctuations within the initial β -Li phase, creating local Li-poor regions (< 23 at.%). Once this thermodynamic condition is met, these regions undergo a martensitic transformation, generating HCP variants. This process can be supported by the updated TEM data of the HPHT-500 sample (Supplementary Fig. 14). The primary α -Mg regions and Li-rich regions are clearly distinguished by the HAADF-STEM image (the characteristic is consistent with the as-cast sample, Supplementary Fig. 13). A compositional diffusion layer exists between them, with no crystallographic difference observed (Supplementary Fig. 14b-e). More importantly, HCP variants with different orientations have begun to form within the Li-rich regions (Supplementary Fig. 14g-h). As the consumption of primary α -Mg regions, the HCP variants originating from different Li-rich regions interact with each other, ultimately forming the complex TIT network (Supplementary Fig. 14i). This direct observation provides key evidence for the continuous evolution process: "compositional fluctuation \rightarrow local phase transformation \rightarrow nucleation of multiple HCP variants \rightarrow formation of the TIT network."

Moreover, the crystallographic orientation relationship of this martensitic transformation has been confirmed through updated MD simulations (Supplementary Fig. 11). We subjected the initial β -Li region in the Mg-9Li model to tensile deformation along different directions. The results show that at a strain of 6 %, HCP variants begin to appear locally. When the strain reaches 10 %, the initial β -Li phase is almost completely transformed into HCP variants of different orientations, which then

assemble into a TIT structure. Analysis of the crystallographic orientation during the transformation reveals a $[1\bar{1}1]_{\text{BCC}} // [11\bar{2}0]_{\text{HCP}}$, $(110)_{\text{BCC}} // (0001)_{\text{HCP}}$ relationship, consistent with previously reported martensitic transformations in titanium [Ref. R2].

Therefore, this work reveals a "diffusion-assisted, compositional-fluctuation-induced martensitic transformation" mechanism. It differs from the traditional shear mechanism and highlights the prerequisite role of compositional fluctuations in initiating the phase transformation under high pressure. We have updated the detailed description of this martensitic transformation mechanism in Supplementary Note 2.

[R2] Zahiri, A. H., et al. The role of mechanical loading in bcc-hcp phase transition: tension-compression asymmetry and twin formation. *Acta Mater.* **241**, 118377 (2022).

Supplementary Figure 11 | Crystallographic relationship of martensitic transformation. Tensile deformation was performed along the $[1\bar{1}1]_{\text{BCC}}$ direction at

0 K. The blue region represents α -Mg with an HCP structure, the red region represents β -Li with a BCC structure, and the yellow region represents α -Mg that contains SF structure. **a**, BCC structure in Mg-9Li supercell viewed along $[1\bar{1}1]_{\text{BCC}}$. HCP phase begins to nucleate when the strain reaches 5 %. Subsequently, HCP variants with different orientations gradually interact and form twin structures as the strain increased to 6%. Finally, complete TIT structure has been formed as the strain increased to 10%. **b**, The same results also have been viewed along $[11\bar{3}]_{\text{BCC}}$ direction. **c**, The crystallographic relationship of martensitic transformation viewed along different directions.

Supplementary Figure 12 | Martensitic transformation conditions. **a**, Li concentration variation in α -Mg and β -Li phases during MD simulation process. **b**, The variation curve between the energies of HCP and BCC structures dependent on the concentration of Li. The critical Li concentration is 23 at%.

Supplementary Figure 13 | Microstructure of as-cast Mg-9Li alloy. a, SEM image of the interface of primary α -Mg and β -Li in as-cast Mg-9Li alloy. **b**, HAADF-STEM image of the interface structure. **c**, EDS mapping of the β -Li region. **d**, BF-TEM image of the interface structure. **e**, HRTEM image of the interface structure viewed along $[0001]_{\alpha}$. **f**, HRTEM image of the interface structure viewed along $[11\bar{2}0]_{\alpha}$.

Supplementary Figure 14 | Microstructure of HPHT-500 Mg-9Li alloy. **a**, SEM image of the HPHT-500 Mg-9Li alloy. **b**, BF-TEM image of the primary α -Mg region viewed along $[1\bar{2}1\bar{3}]$ axis. **c**, HAADF-STEM image corresponding to the dashed box in **b**. **d**, High-magnification HAADF-STEM image of the diffusion layer. The inset shows the corresponding EDS mapping. **e**, HRTEM image of the diffusion layer viewed along $[1\bar{2}1\bar{3}]$. **f**, BF-TEM image of the Li-rich layer viewed along $[11\bar{2}0]$ axis. **g**, Local magnification image of **f**. **h**, BF-TEM image of the Li-rich layer viewed along $[11\bar{2}0]$ axis of the other HCP variant. **i**, High-magnification BF-TEM image of the interaction region between two Li-rich layers.

Supplementary Figure 15 | APT analysis of as-cast sample. a1-a3, Reconstructed APT volume showing the distribution of Mg and Li atoms. **b1-b2**, APT results showing the distribution of Li atoms defined by 10 at.% and 23 at.% Li iso-surfaces, respectively. **c**, Concentration profiles along the red arrow in **b1**. **d-e**, Concentration profiles corresponding to 10 at.% and 23 at.% Li iso-surfaces, respectively.

Supplementary Fig. 16 | APT analysis of HPHT-800 sample. a1-a3, Reconstructed APT volume showing the distribution of Mg and Li atoms. **b1-b3**, APT results showing the distribution of Li atoms defined by 10 at.% and 23 at.% Li iso-surfaces, respectively. **c-d**, Concentration profiles corresponding to 10 at.% and 23 at.% Li iso-surfaces, respectively.

5. The crystallographic orientation relationships for BCC→HCP transformation are described but lack direct experimental proof. High-resolution diffraction or HRTEM/FFT indexing evidence should be provided to confirm such relationships.

Response: Thank you very much for your suggestion. We fully appreciate that directly confirming the crystallographic relationship of the BCC→HCP transformation through HRTEM is critical. To address this question, we attempt atomic-scale characterization of the phase interface using aberration-corrected scanning transmission electron microscopy (STEM). However, fundamental limitations arise due to the extremely low atomic number of Li ($Z=3$). In high-angle annular dark-field (HAADF-STEM) image, where contrast is approximately proportional to Z^2 , the signal intensity from Li atoms is only about 1/16 of that from Mg atoms ($Z=12$). This makes the Li lattice signal nearly indistinguishable from noise. Furthermore, Li is highly sensitive to electron beam irradiation, readily suffering from atomic displacement under common imaging conditions, which further impedes stable high-resolution imaging [Ref. R3]. This technical challenge is common in light-element characterization. The figures below show supplementary HAADF-STEM images of residual Li particles with various shapes and sizes in the HPHT-800 Mg-9Li alloy (Response Supplementary Figs. 1-3). The results indicate that, regardless of their morphology and location, all Li particles exhibit a lattice identical to the Mg matrix, and this consistency is observed for both the $[2-1-10]$ and $[01-11]$ zone axes. This is clearly a consequence of the fundamental limitations of TEM imaging.

To address your concern regarding the crystallographic relationship as effectively as possible within current technical limitations, we employ MD simulations as a crucial

supplementary approach. The simulations accurately reproduced the martensitic transformation process and have directly extracted the crystallographic orientation of the transformed regions (Supplementary Fig. 11). The results show that the orientation relationship between the newly formed HCP phase and the parent BCC phase is consistent with the classical martensitic transformation crystallography proposed in our manuscript. Although this computational evidence is not a direct experimental observation, it provides self-consistent crystallographic support for the proposed transformation mechanism at the atomic scale. We have further emphasized these simulation results in the revised manuscript, presenting them as valid evidence for the orientation relationship under the current experimental limitations.

We have made the following modifications to the manuscript:

“Meanwhile, the crystallographic orientation relationship similar to that of martensite-like transformation has been confirmed through MD simulations (Supplementary Fig. 11).”

[R3] Li, Y., et al. Atomic structure of sensitive battery materials and interfaces revealed by cryo-electron microscopy. *Science*. **358**, 506-510 (2017).

Response Supplementary Figure 1 | a, HAADF-STEM image of Li-rich particles with

block morphology. The inset is the corresponding EDS mapping. **b-c**, Atomic scale HAADF-STEM images of interface structure between the Li-rich particle and the matrix viewed along $[2\bar{1}10]$ direction. The positions are marked in **a**. **d-e**, Interface structure between the Li-rich particle and the matrix viewed along $[01\bar{1}1]$ direction. **f**, FFT image corresponding to **e**.

Response Supplementary Figure 2 | a, HAADF-STEM image of the Li-rich particle with a rod morphology viewed along $[2\bar{1}10]$ direction. **b-c**, Atomic scale HAADF-STEM images under different focus. There is a slight height difference between the Li-rich particle and the matrix.

Response Supplementary Figure 3 | a, HAADF-STEM image of the Li-rich particles located at TB. b, The corresponding EDS mapping of a. c, Atomic scale HAADF-STEM image of the Li-rich particle located at $\{10\bar{1}1\}$ TB viewed along $[2\bar{1}10]$ direction. d, The corresponding FFT image of the interface structure marked by yellow dashed box in c.

Supplementary Figure 11 | Crystallographic relationship of martensitic transformation. Tensile deformation was performed along the $[1\bar{1}1]_{\text{BCC}}$ direction at 0 K. The blue region represents α -Mg with an HCP structure, the red region represents β -Li with a BCC structure, and the yellow region represents α -Mg that contains SF structure. **a**, BCC structure in Mg-9Li supercell viewed along $[1\bar{1}1]_{\text{BCC}}$. HCP phase begins to nucleate when the strain reaches 5 %. Subsequently, HCP variants with different orientations gradually interact and form twin structures as the strain increased to 6%. Finally, complete TIT structure has been formed as the strain increased to 10%. **b**, The same results also have been viewed along $[113]_{\text{BCC}}$ direction. **c**, The crystallographic relationship of martensitic transformation viewed along different directions.

6. Several twin-boundary HRTEM images are insufficiently resolved to identify the exact atomic configuration. Improved atomic-resolution STEM characterizations are needed to verify the proposed interface structures.

Response: Thank you for your valuable suggestion. We fully agree that clear characterization of the interface at the atomic scale is crucial for verifying the proposed TIT structure. To directly address your concern, we have followed your recommendation and re-imaged the key regions using aberration-corrected scanning transmission electron microscopy (HAADF-STEM). The data has been updated in the Fig. 2, and the corresponding descriptions have also been revised in the manuscript.

Fig. 2 | Microstructures. **a**, BF-STEM image of TIT structures in an HPHT-800 Mg-9Li alloy viewed along $[11\bar{2}0]$. **b**, HAADF-STEM image of a typical i-TIT structure. **c**, HAADF-STEM image of a typical a-TIT structure. The yellow dashed lines and red dashed lines represent $\{10\bar{1}1\}$ TBs and $\{10\bar{1}3\}$ TBs, respectively. **d**, HAADF-STEM image of a typical $\{10\bar{1}1\}$ TB. Li atoms are arranged periodically on the $\{10\bar{1}1\}$ TB. **e-f**, the disclination structures of i-TIT and a-TIT, respectively. The blue solid line represents the Mg matrix plane, and the red solid line represents the $\{10\bar{1}1\}$ TBs. **g**, HRTEM images of basal intrinsic I1-type stacking faults around $\{10\bar{1}1\}$ TBs viewed

along $[11\bar{2}0]$. **h**, The statistical fractions, determined on the basis of TEM images, of various interfaces.

7. Since multiple crystal structures (BCC, FCC, HCP) are mentioned, quantitative elemental distribution data are essential, particularly regarding Li partitioning. The absence of APT/EELS or reliable STEM-EDS analysis is a critical gap that must be addressed.

Response: Thank you for your important suggestion. We have conducted atom probe tomography (APT) experiments as requested and performed a systematic quantitative analysis of the Li distribution.

First, regarding the "FCC ordered structure" mentioned in the manuscript, we would like to clarify that it does not represent an independent, long-range ordered, thermodynamically stable phase. Instead, it refers to metastable regions consisting of only a few atomic layers of an FCC stacking sequence (ABCABC), which form near the triple junctions of the TIT structure. Due to their extremely small size and localized distribution, the current spatial resolution of APT is insufficient for their isolated localization and quantitative analysis. Of course, in order to eliminate its misunderstanding on FCC phase, we have deleted the description related to FCC structure. Therefore, the APT data primarily reflects the compositional information of the BCC and HCP phases.

The updated APT data reveal differences in Li concentration within the HCP variants formed via martensitic transformation, with Li-poor and Li-rich regions showing concentrations of 7.3 at.% and 14.5 at.%, respectively (Supplementary Fig. 16). Additionally, Li-rich nanoparticles similar to those in the β -Li phase of the as-cast sample are uniformly distributed in the HPHT-800 sample (Supplementary Fig. 16 and Supplementary Fig. 13), with a Li concentration of 50 at.%. Furthermore, according to updated first-principles calculations (Supplementary Fig. 12b), the free energies of the BCC and HCP phases intersect at a Li concentration of approximately 23 at.%. This implies that when the local Li concentration exceeds this critical value, the BCC phase is more stable. Conversely, when the concentration falls below this value, the HCP

phase is more stable. Therefore, we believe that the uniformly distributed Li-rich particles is residual BCC structure. This result clearly demonstrates a significant partitioning of Li between the BCC and HCP phases, providing direct compositional evidence for the martensitic transformation related to the diffusion of Li atoms. The relevant APT data has been updated in Supplementary Fig. 15-16.

Supplementary Figure 12 | Martensitic transformation conditions. **a**, Li concentration variation in α -Mg and β -Li phases during MD simulation process. **b**, The variation curve between the energies of HCP and BCC structures dependent on the concentration of Li. The critical Li concentration is 23 at%.

Supplementary Figure 13 | Microstructure of as-cast Mg-9Li alloy. **a**, SEM image of the interface of primary α -Mg and β -Li in as-cast Mg-9Li alloy. **b**, HAADF-STEM

image of the interface structure. **c**, EDS mapping of the β -Li region. **d**, BF-TEM image of the interface structure. **e**, HRTEM image of the interface structure viewed along $[0001]_{\alpha}$. **f**, HRTEM image of the interface structure viewed along $[11\bar{2}0]_{\alpha}$.

Supplementary Figure 15 | APT analysis of as-cast sample. **a1-a3**, Reconstructed APT volume showing the distribution of Mg and Li atoms. **b1-b2**, APT results showing the distribution of Li atoms defined by 10 at.% and 23 at.% Li iso-surfaces, respectively. **c**, Concentration profiles along the red arrow in **b1**. **d-e**, Concentration profiles corresponding to 10 at.% and 23 at.% Li iso-surfaces, respectively.

Supplementary Fig. 16 | APT analysis of HPHT-800 sample. a1-a3, Reconstructed APT volume showing the distribution of Mg and Li atoms. **b1-b3**, APT results showing the distribution of Li atoms defined by 10 at.% and 23 at.% Li iso-surfaces, respectively. **c-d**, Concentration profiles corresponding to 10 at.% and 23 at.% Li iso-surfaces, respectively.

8. If both primary α -Mg and transformation-induced HCP phases are present, their compositional and crystallographic differences must be clarified. Distinguishing these two is necessary to validate the transformation interpretation.

Response: We would like to express our sincere appreciation for your careful reading and kindly comments. Distinguishing between the primary α -Mg and the α -Mg formed via martensitic transformation in the HPHTed sample is difficult. Therefore, we adopted the following approach to address your concern.

First, we attempted to distinguish them from a crystallographic perspective. In the as-cast alloy, the interface between the primary α -Mg and the β -Li phase does not exhibit a strict crystallographic parallel relationship (Supplementary Fig. 13d-f). Furthermore,

distinct Li-rich particles are present within the β -Li phase, whereas such features are absent in the primary α -Mg (Supplementary Fig. 13b-c). This characteristic provides a basis for identifying such interfaces. Subsequently, in the HPHT-800 sample, we attempted to determine the crystallographic relationship between the newly formed HCP variants and the residual BCC phase, aiming to identify the transformation product via the Burgers orientation relationship. However, this approach encountered a fundamental experimental challenge. Due to the extremely low atomic number of Li ($Z=3$), its signal intensity in HAADF-STEM imaging ($\propto Z^2$) is only about 1/16 of that of Mg ($Z=12$). Consequently, despite extensive efforts, all observed Li particle regions exhibited lattice images identical to the HCP-Mg (Response Supplementary Fig. 1-3), preventing direct resolution of their atomic positions for precise crystallographic information. Therefore, we employ MD simulations as a crucial supplementary approach. The simulations accurately reproduced the martensitic transformation process and have directly extracted the crystallographic orientation of the transformed regions (Supplementary Fig. 11). The results show that the orientation relationship between the newly formed HCP phase and the parent BCC phase is consistent with the classical martensitic transformation crystallography proposed in our manuscript. Although this computational evidence is not a direct experimental observation, it provides self-consistent crystallographic support for the proposed transformation mechanism at the atomic scale. In summary, no specific crystallographic orientation relationship exists between the primary α -Mg and β -Li phases. In contrast, the HCP variants formed via martensitic transformation and the β -Li phase maintain the following orientation relationship: $[1\bar{1}1]_{\text{BCC}} // [11\bar{2}0]_{\text{HCP}}$, $(110)_{\text{BCC}} // (0001)_{\text{HCP}}$.

To directly obtain the most crucial evidence of compositional differences, we performed atom probe tomography (APT) experiments. We quantitatively analyzed the composition of the α -Mg/HCP phase in both the as-cast Mg-9Li alloy and the HPHT-800 sample. The updated APT data reveal differences in Li concentration within the HCP variants formed via martensitic transformation, with Li-poor and Li-rich regions showing concentrations of 7.3 at.% and 14.5 at.%, respectively (Supplementary Fig. 16). Moreover, the results indicate that the average Li concentration in the primary α -

Mg phase of the as-cast alloy is 3.3 at.% (Supplementary Fig. 15). This compositional difference serves as the most reliable criterion for distinguishing the two HCP phases. It should be noted that the primary α -Mg phase with the same composition as the as-cast sample has largely disappeared after HPHT treatment. This is due to the large number of Li atoms diffusing into the primary α -Mg during the HPHT process (Supplementary Fig. 14). Therefore, the Li-poor HCP variant may be the primary α -Mg. The relevant APT data has been updated in Supplementary Fig. 15-16.

Supplementary Figure 11 | Crystallographic relationship of Martensitic transformation. Tensile deformation was performed along the $[1\bar{1}1]_{\text{BCC}}$ direction at 0 K. The blue region represents α -Mg with an HCP structure, the red region represents β -Li with a BCC structure, and the yellow region represents α -Mg that contains SF structure. **a**, BCC structure in Mg-9Li supercell viewed along $[1\bar{1}1]_{\text{BCC}}$. HCP phase

begins to nucleate when the strain reaches 5 %. Subsequently, HCP variants with different orientations gradually interact and form twin structures as the strain increased to 6%. Finally, complete TIT structure has been formed as the strain increased to 10%. **b**, The same results also have been viewed along $[113]_{\text{BCC}}$ direction. **c**, The crystallographic relationship of martensitic transformation viewed along different directions.

Supplementary Figure 13 | Microstructure of as-cast Mg-9Li alloy. **a**, SEM image of the interface of primary α -Mg and β -Li in as-cast Mg-9Li alloy. **b**, HAADF-STEM image of the interface structure. **c**, EDS mapping of the β -Li region. **d**, BF-TEM image of the interface structure. **e**, HRTEM image of the interface structure viewed along $[0001]_{\alpha}$. **f**, HRTEM image of the interface structure viewed along $[11\bar{2}0]_{\alpha}$.

Supplementary Figure 14 | Microstructure of HPHT-500 Mg-9Li alloy. **a**, SEM image of the HPHT-500 Mg-9Li alloy. **b**, BF-TEM image of the primary α -Mg region viewed along $[1\bar{2}1\bar{3}]$ axis. **c**, HAADF-STEM image corresponding to the dashed box in **b**. **d**, High-magnification HAADF-STEM image of the diffusion layer. The inset shows the corresponding EDS mapping. **e**, HRTEM image of the diffusion layer viewed along $[1\bar{2}1\bar{3}]$. **f**, BF-TEM image of the Li-rich layer viewed along $[11\bar{2}0]$ axis. **g**, Local magnification image of **f**. **h**, BF-TEM image of the Li-rich layer viewed along $[11\bar{2}0]$ axis of the other HCP variant. **i**, High-magnification BF-TEM image of the interaction region between two Li-rich layers.

Supplementary Figure 15 | APT analysis of as-cast sample. a1-a3, Reconstructed APT volume showing the distribution of Mg and Li atoms. **b1-b2**, APT results showing the distribution of Li atoms defined by 10 at.% and 23 at.% Li iso-surfaces, respectively. **c**, Concentration profiles along the red arrow in **b1**. **d-e**, Concentration profiles corresponding to 10 at.% and 23 at.% Li iso-surfaces, respectively.

Supplementary Fig. 16 | APT analysis of HPHT-800 sample. a1-a3, Reconstructed APT volume showing the distribution of Mg and Li atoms. **b1-b3**, APT results showing the distribution of Li atoms defined by 10 at.% and 23 at.% Li iso-surfaces, respectively. **c-d**, Concentration profiles corresponding to 10 at.% and 23 at.% Li iso-surfaces, respectively.

Response Supplementary Figure 1 | a, HAADF-STEM image of Li-rich particles with block morphology. The inset is the corresponding EDS mapping. **b-c**, Atomic scale HAADF-STEM images of interface structure between the Li-rich particle and the matrix viewed along $[2\bar{1}\bar{1}0]$ direction. The positions are marked in **a**. **d-e**, Interface structure between the Li-rich particle and the matrix viewed along $[01\bar{1}\bar{1}]$ direction. **f**, FFT image corresponding to **e**.

Response Supplementary Figure 2 | **a**, HAADF-STEM image of the Li-rich particle with a rod morphology viewed along $[2\bar{1}10]$ direction. **b-c**, Atomic scale HAADF-STEM images under different focus. There is a slight height difference between the Li-rich particle and the matrix.

Response Supplementary Figure 3 | **a**, HAADF-STEM image of the Li-rich particles located at TB. **b**, The corresponding EDS mapping of **a**. **c**, Atomic scale HAADF-STEM image of the

Li-rich particle located at $\{10\bar{1}1\}$ TB viewed along $[2\bar{1}10]$ direction. **d**, The corresponding FFT image of the interface structure marked by yellow dashed box in **c**.

9. The claim that twin structures inhibit crack initiation and propagation is unsupported because no fracture behavior characterization is provided. Fractography and crack-path analysis are required.

Response: We would appreciate your valuable comments. The strength-ductility mechanism of $\{10-11\}$ - $\{10-11\}$ hierarchical contraction twin and $\{10-13\}$ twins in Mg-Li has been well clarified in our previous works (Ref. R1, R4). Basically, in the case of simple twin boundaries, both ex-situ experiments and molecular simulations results show that the prohibition of dislocation motion can be achieved by TBs. Concurrently, the high ductility can be attained by forming incoherent steps in TBs. Moreover, according to your suggestions, we have supplemented the analysis of tensile fracture morphology and crack propagation paths for the HPHT-800 sample, and the relevant descriptions have been updated to the TIT deformation behaviors section.

We have made the following modifications to the manuscript:

*“Finally, the HPHT-800 sample significantly improves the yield strength without apparent sacrificing its ductility. The possible reasons lie in three aspects. Firstly, as evidenced by TEM observations, the formation of numerous high-stepped BP facets in TBs (**Fig. 6a-c**) can effectively alleviate stress concentration^{14, 28}. Secondly, TBs serve as effective barriers that can significantly facilitate dislocation accommodation (approximately $1.171 \times 10^{14} \text{ m}^{-2}$ in terms of Williamson-Hall (W-H) analysis, **Supplementary Fig. 25 and Fig. 6d**). Thirdly, the high-density TIT structures contains HCP matrices with multiple orientations. This feature can provide favorable crystallographic orientation conditions for the activation of non-basal slip during deformation process. TEM observations confirm the presence of a high density of non-basal $\langle c+a \rangle$ dislocations in the deformed TIT structure, whose density even exceeds that of basal $\langle a \rangle$ dislocations (**Fig. 6e-f**). Consequently, the barrier effect of TBs and the activation of non-basal slip are the primary mechanisms responsible for the high*

work hardening rate, especially under compression condition (**Supplementary Fig. 19**). From the perspective of the fracture surface, the observation of dimple features within grain interiors further supports the plastic deformation mechanism discussed above (**Supplementary Fig. 26**).

[R1] Peng, Q., et al. Structural characteristics of {10-11} contraction twin-twin interaction in magnesium. *Acta Mater.* **192**, 60-66 (2020).

[R4] Peng, Q., et al. Interactive contraction nanotwins-stacking faults strengthening mechanism of Mg alloys. *Acta Mater.* **169**, 36-44 (2019).

Supplementary Figure 26 | Fracture surface and crack propagation. **a**, SEM image of the fracture tip of HPHT-800 Mg-9Li alloy after tensile testing. **b**, High magnification SEM image of GB including crack. **c**, Fracture surface of HPHT-800 Mg-9Li alloys.

10. The manuscript provides extensive characterization and modeling to explain the strength improvement, but offers very limited discussion regarding tensile ductility. In current Mg-Li alloys, achieving ultra-high strength is typically accompanied by a drastic loss of elongation, and mechanical testing is usually limited to compression conditions. In contrast, the authors report relatively good tensile ductility while maintaining high strength, which is highly unusual for this alloy system. A more detailed mechanistic explanation of how ductility is preserved—considering deformation modes, strain hardening behavior, slip/twin activity, and potential detwinning—is necessary to substantiate this result.

Response: Thank you for raising this insightful and critical question. The authors will discuss the underlying ductile mechanism from the perspectives of deformation modes, strain hardening behavior, and the stability of TIT structure.

Firstly, we performed TEM observation and analysis on HPHT-800 sample subjected to tensile fracture (Fig. 6 and Supplementary Fig. 25). The formation of numerous high-stepped BP facets in TBs (Fig. 6a-c) can effectively alleviate stress concentration. This indicates that the TIT structure remained stable during the deformation process and no detwinning occurred. Moreover, the results reveal a high density of dislocations within the TIT structure, where the density of non-basal dislocations ($\langle c \rangle$ or $\langle c+a \rangle$) is even higher than that of basal $\langle a \rangle$ dislocations. This finding is partially consistent with the EBSD analysis, which indicated higher Schmid factors for non-basal slip systems compared to basal slip (Response Supplementary Fig. 4). These dislocations exhibit significant pile-ups and interactions near TIT structures, providing direct evidence that these interfaces can effectively pin dislocation motion. Meanwhile, these accumulated dislocations generate a strong back stress, which continuously contributes to work hardening during deformation, particularly under compression (Supplementary Fig. 19). In addition, to quantitatively characterize the dislocation density, we collected XRD patterns from the tensile-fractured sample and applied the classic Williamson-Hall (W-H) method to analyze the diffraction peak broadening for estimating the total dislocation density. The fitting results indicate that the dislocation density in the deformed material reaches $1.171 \times 10^{14} \text{ m}^{-2}$.

Secondly, in our system, we improve strength by forming TIT network involving a large fraction of TBs, which can effectively pin dislocation motion, analogous to strengthening precipitates. More importantly, these structures can remain stable at room temperature. TEM characterization of the sample aged at room temperature for two years reveals that the fundamental configuration of the TIT network remains intact, with no signs of detwinning, significant coarsening, or fundamental transformation. The hardness decrease (approximately 6%) is primarily attributed to a slight thickening of the {10-11} nanotwins (average thickness ~67 nm), which may be due to the relaxation of internal stress within the TIT structure during prolonged natural aging. Thus, the similar TIT structure can still be well observed even after long-term aging at room temperature. In addition, we further conducted aging experiments at 60 °C. The hardness variation curve indicates that the hardness decreased by 7.6% after 2000 hours. The TEM characterization confirmed that the TIT structure also remained stable at slightly higher temperatures without significant detwinning. These supplementary results demonstrate that the TIT strengthening structure possesses excellent microstructural stability under long-term aging. The relevant data have been updated in Supplementary Fig. 21.

We have made the following modifications to the manuscript:

*“Finally, the HPHT-800 sample significantly improves the yield strength without apparent sacrificing its ductility. The possible reasons lie in three aspects. Firstly, as evidenced by TEM observations, the formation of numerous high-stepped BP facets in TBs (Fig. 6a-c) can effectively alleviate stress concentration^{14, 28}. Secondly, TBs serve as effective barriers that can significantly facilitate dislocation accommodation (approximately $1.171 \times 10^{14} \text{ m}^{-2}$ in terms of Williamson-Hall (W-H) analysis, **Supplementary Fig. 25 and Fig. 6d**). Thirdly, the high-density TIT structures contains HCP matrices with multiple orientations. This feature can provide favorable crystallographic orientation conditions for the activation of non-basal slip during deformation process. TEM observations confirm the presence of a high density of non-basal $\langle c+a \rangle$ dislocations in the deformed TIT structure, whose density even exceeds that of basal $\langle a \rangle$ dislocations (Fig. 6e-f). Consequently, the barrier effect of TBs and*

the activation of non-basal slip are the primary mechanisms responsible for the high work hardening rate, especially under compression condition (**Supplementary Fig. 19**). From the perspective of the fracture surface, the observation of dimple features within grain interiors further supports the plastic deformation mechanism discussed above (**Supplementary Fig. 26**).

Response Supplementary Figure 4 | Schmid factors. a-d, Schmid factors of basal slip and pyramidal slip in tension calculated from IPF image for HPHT-800 Mg-9Li alloy. e-h, Schmid factors of basal slip and pyramidal slip in compression calculated from IPF image for HPHT-800 Mg-9Li alloy.

Fig. 6 | Deformed characteristics. **a**, The morphology of the HPHT-800 Mg-9Li alloy after tensile fracture. **a**, BF-TEM image of the TIT structure. **b-c**, HRTEM image of the deformed i-TIT and a-TIT structure, respectively. **d**, Dual-beam dark-field image when $g=(10\bar{1}1)$. A large number of dislocations have been pinned by TIT structures. **e-f**, Dual-beam dark-field image when $g=(0002)$ and $g=(10\bar{1}1)$, respectively. The results show that a large number of $\langle c \rangle$ and $\langle c+a \rangle$ dislocations have been activated.

Supplementary Figure 21 | Long-term stability of TIT structures. **a**, BF-TEM image of the HPHT-800 Mg-9Li sample aged at ambient after two years. Based on statistical analysis, the average thickness of the nanotwin is 67 nm. **b-c**, HRTEM images of the i-TIT and the a-TIT structures in white dashed box indicated in **a**, respectively. **d**, BF-TEM image of the HPHT-800 Mg-9Li sample aged at 60 °C after 2000 h. Based on statistical analysis, the average thickness of the nanotwin is 78 nm. **e-f**, HRTEM images of the i-TIT and the a-TIT structures in white dashed box indicated in **d**, respectively.

Supplementary Figure 25 | Dislocation density. a , XRD pattern of the HPHT-800 sample after tensile fracture. **b**, The fitted curve for the Classic Williamson-Hall (W-H) method. According to calculated, the dislocation density is approximately $1.171 \times 10^{14} \text{ m}^{-2}$.

11. The claim of two-year aging resistance requires stronger evidence, including microstructural characterization after aging and evaluation at slightly elevated temperatures. The current analysis is insufficient to establish long-term stability.

Response: We thank you for the critical comment. In the case of Mg-Li-based alloys, the strength reason origins from the formation of Li-containing compounds. Owing to the easy decomposition of these structures, the aging of common Mg-Li alloys at room temperature become it's a severe problem. Specifically, the strengthening phases or

precipitates coarsen at low temperature, resulting in the sharp reduce in strength. In our system, we improve strength by forming TIT network involving a large fraction of TBs, which can effectively pin dislocation motion, analogous to strengthening precipitates. More importantly, these structures can remain stable at room temperature. TEM characterization of the sample aged at room temperature for two years reveals that the fundamental configuration of the TIT network remains intact, with no signs of detwinning, significant coarsening, or fundamental transformation. The hardness decrease (approximately 6%) is primarily attributed to a slight thickening of the {10-11} nanotwins (average thickness ~67 nm), which may be due to the relaxation of internal stress within the TIT structure during prolonged natural aging. Thus, the similar TIT structure can still be well observed even after long-term aging at room temperature. In this case, it demonstrates that the TIT structure is more stable than traditional strengthening precipitates.

In addition, we further conducted aging experiments at 60 °C. The hardness variation curve indicates that the hardness decreased by 7.6% after 2000 hours. The TEM characterization confirmed that the TIT structure also remained stable at slightly higher temperatures without significant detwinning. These supplementary results demonstrate that the TIT strengthening structure possesses excellent microstructural stability under long-term aging. The relevant data and descriptions have been updated in Supplementary Fig. 21 and in the TIT deformation behaviour section, respectively.

We have made the following modifications to the manuscript:

“Moreover, even aged at 60 °C for 2000 h, a high hardness value (~136 HV) is detected (Fig. 5e). TEM characterization results demonstrate that the TIT strengthening structure possesses excellent structural stability under both long-term ambient and slightly elevated temperature conditions (Supplementary Fig. 21 and Supplementary Note 4). Therefore, compared to the traditional precipitation strengthening, TIT structure strengthening strategy offers a completely new design concept and mechanism for achieving Mg-Li based alloys with both high strength and excellent long-term stability.”

Supplementary Figure 21 | Long-term stability of TIT structures. **a**, BF-TEM image of the HPHT-800 Mg-9Li sample aged at ambient after two years. Based on statistical analysis, the average thickness of the nanotwin is 67 nm. **b-c**, HRTEM images of the i-TIT and the a-TIT structures in white dashed box indicated in **a**, respectively. **d**, BF-TEM image of the HPHT-800 Mg-9Li sample aged at 60 °C after 2000 h. Based on statistical analysis, the average thickness of the nanotwin is 78 nm. **e-f**, HRTEM images of the i-TIT and the a-TIT structures in white dashed box indicated in **d**, respectively.

12. Considering the HPHT temperature approaching ~800°C, possible changes in alloy composition, especially Li evaporation or burn loss, must be evaluated and reported.

Response: Thank you for your thoughtful comment. On the one hand, the HPHT is firstly a closed-well condition. If it is not airtight condition, the setup might be destroyed, even explode under solid-liquid samples due to the volume changes. As a result, the Mg or Li can not burn and the loss is ignored. On the other hand, it is worth noting that the change in phase transformation temperature (T_{eq}) caused by pressure change can

be calculated by Clausius-Clapeyron equation [Ref. R5]:

$$T_{eq}(P_0 + \Delta P) = T_{eq}(P_0) + \Delta P \frac{dT_{eq}}{dp} = T_{eq,0} + \Delta P \frac{T_{eq,0} \cdot \Delta v}{l_{1 \rightarrow 2}}$$

where $T_{eq,0}$ is the equilibrium temperature of the phase transition, dT_{eq} is the change in melting temperature, P_0 (Pa) is the pressure and ΔP (Pa) is the change in pressure, $l_{1 \rightarrow 2}$ is specific enthalpy of phase transformation and Δv is the corresponding specific volume change. The coefficient of dT_{eq}/dP of Mg is calculated as 49.9 K/GPa based on the previous results of Mg under high pressure [Ref. R6]. By substituting the dT_{eq}/dP values, the melting point of metallic Mg under 6 GPa is about 1236 K (963 °C). Therefore, the HPHTed of Mg-Li alloy under 6GPa at 800 °C is a pure solid phase. The loss of alloy is expected to be very low. To verify these results, we also test the alloy composition of as-cast and HPHT-800 Mg-9Li alloy using ICP. The results show that the composition did not undergo significant changes before and after the HPHT treatment. The ICP results are as shown in Supplementary Table 1. The relevant descriptions have also been updated in methods section.

We have made the following modifications to the manuscript:

“Actual chemical compositions (wt. %) of as-cast and HPHT-800 Mg-9Li alloy determined by inductively coupled plasma mass spectrometry (ICP-OES) are listed in Supplementary Table 1.”

[R5] Sobczak J J., et al. Effect of pressure on solidification of metallic materials. *Int. J. Cast Metal. Res.* **25**, 1-14 (2012).

[R6] Fu H., et al. Achieving synergistic strength-ductility optimization in ultralight Mg-11Li alloy by ultrahigh pressure treatment. *Materials Characterization.* **225**, 115162 (2025).

Supplementary Table 1. Actual chemical compositions (wt. %) of as-cast and HPHT-800 Mg-9Li alloys by ICP-OES.

Alloy (wt.%)	Mg	Li	Fe	Ni	Cu	Bal.
-----------------	----	----	----	----	----	------

As-cast	91.11	8.87	0.005	0.001	0.002	0.012
HPHT- 800	91.34	8.63	0.005	0.002	0.003	0.02

13. The manuscript reports much higher compressive strength than tensile strength, yet Mg-based alloys are known for strong tension-compression asymmetry. Discussion on this asymmetry and the potential need for in situ tensile testing is required.

Response: Thank you very much for your constructive suggestions. First, it is necessary for us to clarify that the mechanical properties, especial the micro-sized samples, are greatly affected by the sizes (as evidenced by the nanopillars). Thus, it is not accurate to compare the tension-compression asymmetry between micro-sized compression and macro-sized tensile tests. Therefore, we test the macro-sized compression performance, and the relative results are added in the Supplementary Fig. 19. Moreover, we have added a description of the compression yield strength in the manuscript.

We have made the following modifications to the manuscript:

“Moreover, the compression yield strength is up to 534 MPa (Supplementary Fig. 19).”

Based on these results, we can find that the tension-compression asymmetry is abnormal compared to common Mg alloys. Next, we will discuss this abnormal behavior. The tension-compression yield asymmetry describes the difference in mechanical response under tension and compression, usually induced by {10-12} deformation twinning and texture from thermomechanical processing (compressive yield/tensile yield < 1) [Ref. R7]. Theoretically, texture weakening and twinning suppression are two effective strategies to reduce or eliminate this asymmetry [Ref. R7]. We observed that the compression strength of the HPHT-800 sample exceeds its tension strength, which differs from the tension-compression yield asymmetry typically exhibited by conventional Mg alloys. We believe this reflects the unique deformation characteristics of the distinct microstructure constructed by the high density TIT

network in this work. This "abnormal" tension-compression yield asymmetry primarily stems from the following two aspects:

Firstly, the TIT structure itself strongly inhibits twin initiation. The TIT network divides the HCP matrix into numerous nano-scale "domains." This extreme grain refinement significantly raises the stress threshold for twin nucleation [Ref. R8]. Consequently, during the deformation process, plastic strain is primarily accommodated by dislocation slip, the premature compressive yield caused by the rapid initiation and growth of twins was avoided.

Secondly, the HPHT-800 sample exhibits no distinct texture (Supplementary Fig. 1). The nearly random grain orientation distribution renders the material macroscopically isotropic, leading to similar Schmid factors for the activation of basal and non-basal dislocation slip systems under tensile and compressive conditions (Response Supplementary Fig. 4). This eliminates the intrinsic tension-compression asymmetry caused by strong texture.

In summary, the higher compressive strength can likely be attributed to the following: Under compressive loading, the TIT structure acts as three-dimensional interlocking network obstacles, which can more effectively hinder dislocation motion and induce stronger non-basal slip and dislocation multiplication, thereby resulting in a higher work-hardening rate (Supplementary Fig. 19c). Under tensile loading, although the strengthening mechanisms remain active, the Schmid factors for basal and non-basal slip are slightly higher than in compression, probably leading to an earlier entry into the plastic deformation stage. Therefore, the tensile yield strength is slightly lower than the compressive yield strength.

[R7] Yin, D. D., et al. Tension-compression asymmetry and the underlying slip/twinning activity in extruded Mg-Y sheets. *Int. J. Plasticity*. **136**, 102878 (2021).

[R8] Dogan, E., et al. Reduction in tension-compression asymmetry via grain refinement and texture design in Mg-3Al-1Zn sheets. *Mater. Sci. Eng. A* **610**, 220-227 (2014).

Supplementary Figure 19 | Vickers hardness and compression properties. a, Variations in Vickers hardness of Mg-9Li alloys as a function of high-pressure and high-temperature treatment. **b,** Variations in Vickers hardness of HPHT-800 Mg-9Li alloys as a function of holding time. **c,** Compressive true stress–strain curves of Mg-9Li alloys in various states at a nominal strain rate of 10^{-3} s^{-1} .

Response Supplementary Figure 4 | Schmid factors. a-d, Schmid factors of basal slip and pyramidal slip in tension calculated from IPF image for HPHT-800 Mg-9Li alloy. **e-h,** Schmid factors of basal slip and pyramidal slip in compression calculated from

IPF image for HPHT-800 Mg-9Li alloy.

14. Dislocation behavior, including type and density, is not analyzed despite its importance in twin-induced strengthening. TEM-based identification and XRD-based density estimation should be included.

Response: Thank you for your important suggestion. We fully appreciate that analyzing dislocation behavior during deformation is crucial for understanding the strengthening mechanism of the TIT structure. According to your suggestion, we have supplemented our analysis with TEM-based dislocation characterization and XRD-based dislocation density estimation.

On the one hand, we performed TEM observation and analysis on HPHT-800 sample subjected to tensile fracture. The results reveal a high density of dislocations within the TIT structure, where the density of non-basal dislocations ($\langle c \rangle$ or $\langle c+a \rangle$) is even higher than that of basal $\langle a \rangle$ dislocations. This finding is partially consistent with the EBSD analysis, which indicated higher Schmid factors for non-basal slip systems compared to basal slip (Response Supplementary Fig. 4). These dislocations exhibit significant pile-ups and interactions near TIT structures, providing direct evidence that these interfaces can effectively pin dislocation motion.

In addition, to quantitatively characterize the dislocation density, we collected XRD patterns from the tensile-fractured sample and applied the classic Williamson-Hall (W-H) method to analyze the diffraction peak broadening for estimating the total dislocation density. The fitting results indicate that the dislocation density in the deformed material reaches $1.171 \times 10^{14} \text{ m}^{-2}$. The supplementary TEM data and the quantitative XRD data have been updated in Fig. 6 and Supplementary Fig. 25. The corresponding descriptions have also been updated in the manuscript.

We have made the following modifications to the manuscript:

“Finally, the HPHT-800 sample significantly improves the yield strength without apparent sacrificing its ductility. The possible reasons lie in three aspects. Firstly, as evidenced by TEM observations, the formation of numerous high-stepped BP facets

in TBs (**Fig. 6a-c**) can effectively alleviate stress concentration ^{14, 28}. Secondly, TBs serve as effective barriers that can significantly facilitate dislocation accommodation (approximately $1.171 \times 10^{14} \text{ m}^{-2}$ in terms of Williamson-Hall (W-H) analysis, **Supplementary Fig. 25 and Fig. 6d**). Thirdly, the high-density TIT structures contains HCP matrices with multiple orientations. This feature can provide favorable crystallographic orientation conditions for the activation of non-basal slip during deformation process. TEM observations confirm the presence of a high density of non-basal $\langle c+a \rangle$ dislocations in the deformed TIT structure, whose density even exceeds that of basal $\langle a \rangle$ dislocations (**Fig. 6e-f**). Consequently, the barrier effect of TBs and the activation of non-basal slip are the primary mechanisms responsible for the high work hardening rate, especially under compression condition (**Supplementary Fig. 19**). From the perspective of the fracture surface, the observation of dimple features within grain interiors further supports the plastic deformation mechanism discussed above (**Supplementary Fig. 26**).

Response Supplementary Figure 4 | Schmid factors. a-d, Schmid factors of basal slip and pyramidal slip in tension calculated from IPF image for HPHT-800 Mg-9Li alloy. e-h, Schmid factors of basal slip and pyramidal slip in compression calculated from IPF image for HPHT-800 Mg-9Li alloy.

Fig. 6 | Deformed characteristics. **a**, The morphology of the HPHT-800 Mg-9Li alloy after tensile fracture. **a**, BF-TEM image of the TIT structure. **b-c**, HRTEM image of the deformed i-TIT and a-TIT structure, respectively. **d**, Dual-beam dark-field image when $g=(10\bar{1}1)$. A large number of dislocations have been pinned by TIT structures. **e-f**, Dual-beam dark-field image when $g=(0002)$ and $g=(10\bar{1}1)$, respectively. The results show that a large number of $\langle c \rangle$ and $\langle c+a \rangle$ dislocations have been activated.

Supplementary Figure 25 | Dislocation density. a , XRD pattern of the HPHT-800 sample after tensile fracture. **b**, The fitted curve for the Classic Williamson-Hall (W-H) method. According to calculated, the dislocation density is approximately $1.171 \times 10^{14} \text{ m}^{-2}$.

15. The lack of Li characterization throughout the paper is a major issue because Li plays a critical role in Mg-Li alloys. Proper detection and mapping of Li are necessary to support any structural or mechanistic claims.

Response: Thank you for raising this critical point. We fully agree that chemical characterization of Li is essential for substantiating the microstructural evolution and mechanisms proposed in our study. To address this directly, we have conducted complementary Atom Probe Tomography (APT) experiments to obtain quantitative composition data for the key phases.

The APT analysis was performed on both the as-cast Mg-9Li alloy and the HPHT-800 Mg-9Li alloy. In the as-cast sample, the measured average Li concentrations are 3.3 at.% for the primary α -Mg phase and 20.2 at.% for the β -Li phase. In the HPHT-800 sample, the analysis of distinct regions reveals an average Li concentration of 7.3 at.% for the Li-poor HCP variants and 14.5 at.% for Li-rich HCP variants. Additionally, Li-rich nanoparticles similar to those in the β -Li phase of the as-cast sample are uniformly distributed in the HPHT-800 sample, with a Li concentration of 50 at.%.

This quantitative data provides direct evidence of significant Li partitioning between the different phases. Most importantly, the distinct Li concentration measured for the HCP phase in the HPHT sample (markedly different from that of the primary α -Mg in the as-cast alloy) serves as a definitive compositional identification. This difference strongly supports our interpretation of its origin via a compositionally-driven phase transformation, rather than it being remnant primary α -Mg. The complete APT data have been provided in Supplementary Figs. 15-16.

Supplementary Figure 15 | APT analysis of as-cast sample. a1-a3, Reconstructed APT volume showing the distribution of Mg and Li atoms. **b1-b2**, APT results showing the distribution of Li atoms defined by 10 at.% and 23 at.% Li iso-surfaces, respectively. **c**, Concentration profiles along the red arrow in **b1**. **d-e**, Concentration profiles corresponding to 10 at.% and 23 at.% Li iso-surfaces, respectively.

Supplementary Fig. 16 | APT analysis of HPHT-800 sample. a1-a3, Reconstructed APT volume showing the distribution of Mg and Li atoms. **b1-b3**, APT results showing the distribution of Li atoms defined by 10 at.% and 23 at.% Li iso-surfaces, respectively. **c-d**, Concentration profiles corresponding to 10 at.% and 23 at.% Li iso-surfaces, respectively.

16. The effect of transformation on density and elastic modulus should be considered, since these properties are crucial for Mg-Li alloys. Direct measurements before and after HPHT are needed.

Response: Thank you for your important suggestion. To directly address your question, we have supplemented the analysis of density and elastic modulus changes in the alloy before and after HPHT treatment. The details are as follows:

Firstly, to measure density accurately, cylindrical specimens of identical dimensions (diameter: 7.78 mm, height: 4.8 mm) were prepared from both the as-cast and HPHT-800 samples using diamond wire cutting to minimize machining damage and thermal effects. Mass was measured using a precision electronic balance, and volume was calculated from the geometric dimensions (Response Supplementary Fig. 5). After calculation, the density of the as-cast sample is approximately 1.4418 g/cm³, and the density of the HPHT-800 sample is approximately 1.44197 g/cm³. The results show no significant change in density after HPHT treatment. This finding is consistent with the compositional analysis (Supplementary Table 1), which indicates the overall alloy composition remained stable before and after HPHT treatment.

Secondly, the nanoindentation technique was employed to quantitatively characterize the elastic modulus. For the as-cast sample, four test points were selected in clear regions of the α -Mg phase and the β -Li phase, respectively, and the average value was taken as the elastic modulus of the as-cast alloy. For the HPHT-800 sample with a homogeneous microstructure, nine randomly selected points on the surface were measured for statistical analysis. The results indicate that the elastic modulus of the sample after HPHT treatment (50.07 GPa) is slightly higher than that of the as-cast

sample (47.67 GPa) (Supplementary Fig. 20). This change is primarily attributed to the reduced volume fraction of the β -Li phase with lower modulus after HPHT treatment. In fact, the modulus of the α -Mg regions in as-cast sample is close to the overall modulus of the sample after HPHT treatment, further supporting this explanation.

Response Supplementary Figure 5 | The quality of the as-cast sample and the HPHT-800 sample. They have the same volume.

Supplementary Figure 20 | Elastic modulus of as-cast and HPHT-800 samples.

a-b, The nanoindentation load-depth variation curves of the α -Mg and β -Li phases in the as-cast Mg-9Li sample respectively. **c,** The nanoindentation load-depth variation curves of the HPHT-800 Mg-9Li sample. **d,** The elastic modulus of as-cast and HPHT-800 samples obtained from the load-indentation depth curves using the Oliver-Pharr method.

17. The transformation-induced HCP structure may represent a metastable supersaturated solid solution given the alloy's phase diagram position. Thermodynamic reasoning supporting its long-term stability is needed.

Response: Thank you for your beneficial suggestion. The free energy of a supersaturated solid solution is higher than that of its equilibrium state. Therefore, from a thermodynamic perspective, it has a tendency to decompose spontaneously to lower the system's free energy. However, although a supersaturated solid solution is thermodynamically unstable, an energy barrier must be overcome for it to transform into a more stable equilibrium state. At room temperature, this transformation process may be extremely slow, to the extent that it is virtually undetectable within the

observation time. We will discuss this from the following aspects.

First, from a thermodynamic perspective, the chemical free energy (G_{chem}) of a supersaturated solid solution is higher than that of the equilibrium state, resulting in a negative driving force for decomposition ($\Delta G_{\text{chem}} < 0$). Considering chemical free energy alone, the HCP structure is theoretically unstable and would spontaneously decompose towards equilibrium. However, the martensitic transformation does not produce isolated HCP structures but forms an interlocked network with triple twin junction nodes as the core. Consequently, the stability of this network is governed by the total Gibbs free energy (G_{total}):

$$G_{\text{total}} = G_{\text{chem}} + G_{\text{interface}} = G_{\text{chem}} + \sum_i \gamma_i A_i \quad (1)$$

where G_{chem} is the chemical free energy, $G_{\text{interface}}$ is the total interfacial energy, γ_i is the interfacial energy of the i -th type of interface, and A_i is the area of that interface. The interfaces between the HCP structure generated by martensitic transformation and the residual BCC parent phase are typically coherent or semi-coherent (with relatively low γ_{phase}). However, according to XRD refinement results, the volume fraction of β -Li in the HPHT-800 sample is very small, meaning A_{phase} is minimal. More importantly, the TIT structure contains a high density of $\{10\bar{1}1\}$ twins, whose boundaries are low-energy coherent interfaces in Mg alloys ($\gamma_{\text{twin}} \approx 84 \text{ mJ/m}^2$). After martensitic transformation, the system enters a state composed of an extremely high density of low-energy interfaces (large A_{twin} , small γ_{twin}). For this structure to destabilize (e.g., through decomposition of the HCP structure or transformation into equilibrium phases), new phase interfaces with higher energy would have to be continuously created, thereby increasing $G_{\text{interface}}$. Furthermore, the growth and assembly of β -Li particles would inevitably involve interface reconstruction, transforming the existing low-energy coherent TBs into high-energy large-angle β -Li GBs. This would cause a sharp increase in $G_{\text{interface}}$, creating a substantial energy barrier that cannot be overcome at room temperature or even slightly elevated temperatures. This is supported by the TEM results that the fundamental configuration of the TIT structure remains intact, with no signs of detwinning, significant coarsening, or transformation observed after aging

at room temperature and slightly elevated temperatures (Supplementary Fig. 21).

Second, even if the system could overcome the aforementioned thermodynamic barrier and nucleate β -Li phase within the HCP supersaturated solid solution, its growth would require long-range diffusion of Li atoms through the HCP structure to reach the nucleation of β -Li. From a kinetic perspective, this diffusion process is governed by Fick's laws and thermal activation, with the diffusion coefficient D given by:

$$D = D_0 \exp(-Q/k_B T) \quad (2)$$

where D_0 is the pre-exponential factor, Q is the activation energy for diffusion, k_B is the Boltzmann constant, and T is the absolute temperature. Crucially, the extensive TIT structure generated by martensitic transformation significantly complicates the diffusion paths for Li atoms, effectively increasing the effective activation energy (Q_{eff}). Additionally, the periodic arrangement of Li atoms along $\{10\bar{1}1\}$ TBs is clearly observable (Fig. 2d), indicating that a fraction of Li atoms are pinned at these boundaries after the martensitic transformation. Therefore, at room temperature (low T), with a high Q_{eff} and the pinning effect at TBs, the diffusion coefficient D becomes exceedingly small. This means Li atoms cannot achieve the compositional redistribution required for the decomposition of the HCP supersaturated solid solution. In summary, we have analyzed and explained the potential long-term stability of the HCP supersaturated solid solution from both thermodynamic and kinetic perspectives. This discussion has been updated to Supplementary Note 4.

Supplementary Figure 21 | Long-term stability of TIT structures. **a**, BF-TEM image of the HPHT-800 Mg-9Li sample aged at ambient after two years. Based on statistical analysis, the average thickness of the nanotwin is 67 nm. **b-c**, HRTEM images of the i-TIT and the a-TIT structures in white dashed box indicated in **a**, respectively. **d**, BF-TEM image of the HPHT-800 Mg-9Li sample aged at 60 °C after 2000 h. Based on statistical analysis, the average thickness of the nanotwin is 78 nm. **e-f**, HRTEM images of the i-TIT and the a-TIT structures in white dashed box indicated in **d**, respectively.

18. The FCC-ordered structure mentioned is neither clearly identified nor supported by XRD evidence. Detailed crystallographic characterization and explanation of its formation are needed.

Response: We would appreciate your valuable comments. First, it is essential to clarify that the "FCC ordered structure" described in our manuscript is not an independent, long-range ordered thermodynamically stable phase, and its size and quantity are relatively small. It should be noted that the detection volume of XRD is

much larger than the field of view in TEM, and its signal represents the statistical average of all structures within the irradiated area. Since the metastable FCC region is only a few nanometers and is not a stable phase, its weak diffraction signal will be completely masked by the strong diffraction background noise generated by the dominant HCP phase. Consequently, they cannot be distinguished in the XRD pattern. Their existence is primarily confirmed by MD simulations (Figs. 4d1-d4), and the characteristic is several atomic layer metastable regions with an FCC stacking sequence (ABCABC) near the TIT structure. Therefore, to avoid misunderstanding, we have deleted the description about the FCC structure.

19. The claimed reduction of anisotropy requires experimental proof through mechanical testing in multiple orientations or directions. Current evidence is insufficient.

Response: Thank you for your valuable feedback on our manuscript. We fully acknowledge that verifying the reduction in material anisotropy requires macroscopic mechanical testing along multiple directions as direct evidence. However, unlike processes such as rolling or extrusion, the HPHT process does not possess the strong directionality. As shown in our supplementary EBSD analysis, the HPHT-800 sample does not exhibit a distinct texture (Supplementary Fig. 1). Therefore, our mechanical testing already includes a large number of randomly oriented grains, effectively representing multi-directional assessment. To further support this point, we have updated one additional stress-strain curve in the Fig. 5a to illustrate the material's good isotropic behavior.

Furthermore, to directly address the key question of whether the mechanical response of the TIT structure itself exhibits significant anisotropy, we performed molecular dynamics (MD) simulations. We constructed atomic models of typical i-TIT, a-TIT, and conventional grain boundary (GB) structures and systematically applied loading along different directions. The simulation results (Fig. 5f and Supplementary Fig. 23) clearly show that two TIT structures exhibit highly similar yielding behaviors under different loading directions, with the mechanical response of the i-TIT structure being

particularly uniform. In contrast, the conventional GB model shows clearly distinct strength responses depending on the loading direction. This comparison strongly demonstrates that the TIT unit composed of interlocked twin boundaries effectively homogenizes the resistance to dislocation nucleation and motion across different directions. This provides a key explanation for our claim that the TIT structure helps reduce the macroscopic anisotropy of the material.

Supplementary Figure 1 | Microstructure of Mg-9Li samples. **a**, Surface morphology and dimension of HPHT samples. The diameter and height are 30 mm and 15 mm, respectively. **b-c**, IPF image and the distribution of β -Li phase (blue) of the as-cast sample. **d-e**, Optical structure, IPF image, and pole figures of the HPHT-800 sample. **f**, Grain size variation of the different HPHT Mg-9Li samples.

Fig. 5a | Mechanical properties of the different sample.

20. The aging resistance in this work should not be directly compared with precipitation-strengthened Mg-Li alloys, because the underlying aging mechanisms are fundamentally different. For binary Mg-10Li alloys, no aging softening typically occurs from a mechanistic perspective. However, the alloy studied in this manuscript exhibits a certain degree of softening (~6%), which contradicts the statement of excellent aging resistance. The validity of this performance claim therefore needs further justification.

Response: We thank you for the critical comment. However, we respectfully disagree with the expression.

As the reviewer mentioned, the ageing softening just occurs in some traditional Mg-Li-X alloys, wherein the containing Li phase readily coarsening or decomposes during the nature aging at room or low temperatures. It demonstrates that the common precipitate-strengthening is not suitable for Mg-Li alloys. On the contrary, due to the formation of high stability of new TIT structure, the high values (~150 HV) can be well remained, which is far high than that of the as-cast Mg-Li sample (~30 HV) correspondingly. Based on these results, two aspects can be inferred. Firstly, differing from precipitate-strengthening in Mg-Li-X samples, TIT structure is very stable.

Secondly, there is no precipitate-strengthening in Mg-Li binary alloys, in which it is strengthening by grain refining or solid solution. However, due to the induction of TIT structures in Mg-Li binary alloys, the strength is remarkably improved, demonstrating the strengthening role is very effective.

In addition, as the reviewer's mentioned, there is a slight reduction (6% for two years) in the HPHTed Mg-Li alloys. The "excellent aging resistance" we mentioned in the manuscript refers to structural stability. The TIT structure produced via martensitic transformation maintains fundamental configuration at both ambient temperature and slightly elevated temperatures, without signs of detwinning, significant coarsening, or a change in type (Supplementary Fig. 21). Although a slight strength loss (~46.71 MPa) occurs due to stress relaxation, the alloy retains a strength approximately nine times higher than its as-cast even after two years. Compared to the significant performance degradation that may occur in many traditional alloys after aging, the stability of the overall performance platform of our alloy is notable. Regarding the statement you made about the binary Mg-10Li alloy not typically exhibiting age-hardening phenomenon, we acknowledge the validity of your viewpoint. However, compared to the traditional precipitation strengthening pathway, the study aims to emphasize that the novel TIT structure strengthening strategy offers a completely new design concept and mechanism for achieving Mg-Li based alloys with both high strength and excellent long-term stability. The relevant description has been updated in the manuscript.

We have made the following modifications to the manuscript:

“Moreover, even aged at 60 °C for 2000 h, a high hardness value (~136 HV) is detected (Fig. 5e). TEM characterization results demonstrate that the TIT strengthening structure possesses excellent structural stability under both long-term ambient and slightly elevated temperature conditions (Supplementary Fig. 21 and Supplementary Note 4). Therefore, compared to the traditional precipitation strengthening, TIT structure strengthening strategy offers a completely new design concept and mechanism for achieving Mg-Li based alloys with both high strength and excellent long-term stability.”

Supplementary Figure 21 | Long-term stability of TIT structures. **a**, BF-TEM image of the HPHT-800 Mg-9Li sample aged at ambient after two years. Based on statistical analysis, the average thickness of the nanotwin is 67 nm. **b-c**, HRTEM images of the i-TIT and the a-TIT structures in white dashed box indicated in **a**, respectively. **d**, BF-TEM image of the HPHT-800 Mg-9Li sample aged at 60 °C after 2000 h. Based on statistical analysis, the average thickness of the nanotwin is 78 nm. **e-f**, HRTEM images of the i-TIT and the a-TIT structures in white dashed box indicated in **d**, respectively.

21. The in situ synchrotron experiment conditions appear different from actual HPHT processing. The authors should explain how well the in situ observations represent the real processing conditions and microstructure evolution.

Response: We thank you for the critical comment. As the reviewer's mentioned, the experimental conditions are slightly different. The main difference originates from the different sample dimensions, different stress-loading modes and different heating modes. Firstly, in the case of the in-situ synchrotron experiment, the size of sample is smaller (~250 μm diameter). In contrast, the diameter of HPHTed sample is about 30 mm. Secondly, the in-situ synchrotron experiments employed a diamond anvil cell (DAC), which operates under uniaxial compression. Pressure is applied uniaxially to the sample through two opposing diamond anvils. In contrast, the six-anvil press used for HPHT processing generates multi-axial (quasi-hydrostatic) compression. Pressure is applied simultaneously and synchronously from three orthogonal directions in three-dimensional space via six independently driven anvils. Consequently, the stress fields imposed on the sample by these two apparatuses are different, which may lead to shifts in specific parameters. Thirdly, in the case of in-situ synchrotron experiment, the temperature is heated by laser light. On the contrary, the HPHTed samples are heated by resistance wire (control the power). Therefore, the critical temperature of phase transformation is slightly different between two methods. Strictly speaking, the experimental conditions of in-situ synchrotron tests are more accurate, which is the main reason we performed. However, in the case of the macro-sized samples, we generally need increase the surface temperature of samples to form a temperature gradient, which can ensure the TIT structure occurs in the whole sample, especially in the core of the samples. This is also the reasons for the different critical temperature value.

It is well known that the phase transition of materials under specific thermodynamic conditions (pressure and temperature) is an inherent property of the materials. The objective of our in-situ synchrotron study was not to replicate the HPHT process but to unveil its physical essence. We simplified the complex stress field of the HPHT process to focus on directly observing the intrinsic transformation laws, such as $\beta\text{-Li}$

(BCC) \rightarrow α -Mg (HCP) evolution in Mg-Li alloys, under precisely controlled extreme hydrostatic pressure and high temperature. During the actual HPHT process, as long as similar thermodynamic conditions are locally reached within the material, the same intrinsic transformation mechanism is activated. Therefore, the microstructural evolution laws revealed by the in-situ experiments are highly comparable and consistent with the evolution process activated during the actual HPHT processing, and they accurately reflect the underlying mechanism of the latter.

Reply to the report of referee 2

1. Fig 1.b: please zoom in on one or 2 grains to clarify you schematic.

Response: Thank you for your valuable advice. We have magnified one of grains and the Fig. 1 has been updated.

2. Please explain more carefully your initial microstructure as well as the HPHT one in terms of phase proportions, crystallographic texture, grain size, phase morphology ... The role of the Li-rich BCC phase seems to be key in the TIT processed but the localization of this minor phase within the matrix is not clear. Also how do the authors explain the presence of TIT within HCP regions that do not face phase transformation? Please clarify that in the manuscript. Without that, the importance of manufacturing a

duplex phase alloy cannot be understood clearly.

Response: Thank you for your beneficial suggestion.

Firstly, we have updated the EBSD data for both the as-cast and HPHT-800 samples (Supplementary Fig. 1). The as-cast sample consists of α -Mg and β -Li phases, with the α -Mg exhibiting a lamellar morphology and the β -Li appearing as island-like structures. Statistical analysis shows that the grain size of the as-cast sample is approximately 530 μm . It should be noted that quantitative phase fraction analysis via EBSD is subject to limitations due to the field of view, which may introduce errors. According to XRD refinement results, the phase fractions of α -Mg and β -Li in the as-cast sample are 45% and 55%, respectively (Supplementary Fig. 8). Based on the updated APT results, the Li concentrations in the α -Mg and β -Li phases are 3.3 at.% and 20.2 at.% (local area is over 50%), respectively (Supplementary Fig. 15). The Li concentration at the interface between the α and β phases is approximately 10 at.%. Additionally, Li-rich nanoparticles are present within the β -Li phase but are absent in the α -Mg matrix, which is consistent with the TEM observations (Supplementary Fig. 13). According to the 23 at.% Li iso-surface analysis, the concentration of Li-rich nanoparticles is approximately 50 at.%. The HPHT-800 sample also consists of two phases. However, the phase fraction of β -Li has decreased to approximately 10% (Supplementary Fig. 8), and the grain size has been refined to 94 μm . EBSD results indicate that the HPHT-800 sample lacks a distinct texture. It is important to note that the relatively low indexing rate for this sample is primarily attributed to the high internal stresses induced by the high density of nanotwins. The updated APT data further reveal differences in Li concentration within the HCP variants formed via martensitic transformation, with Li-poor and Li-rich regions showing concentrations of 7.3 at.% and 14.5 at.%, respectively (Supplementary Fig. 16). Additionally, Li-rich nanoparticles similar to those in the β -Li phase of the as-cast sample are uniformly distributed in the HPHT-800 sample, with a Li concentration of 50 at.%. Notably, the primary α -Mg phase has almost disappeared after HPHT treatment, which is attributed to significant Li diffusion into the primary α -Mg during HPHT treatment (Supplementary Fig. 14).

Secondly, regarding the viewpoint of reviewer that “the role of the Li-rich BCC phase seems to be key in the TIT processed but the localization of this minor phase within the matrix is not clear,” the updated EBSD and APT data provide clear localization of the β -Li phase and the distribution of Li atoms.

Finally, regarding the statement of reviewer’s mention that “how do the authors explain the presence of TIT within HCP regions that do not face phase transformation”, the author believe the reviewer may have some misunderstandings. We would like to provide a detailed explanation of the formation process of the TIT structure. The BCC→HCP microstructure evolution observed in this work is indeed a martensitic transformation driven by compositional fluctuations. The specific conditions and mechanisms are as follows:

Unlike conventional martensitic transformations driven directly by shear at low temperatures or under stress, the transformation in this study is triggered by the diffusion and redistribution of Li atoms during the HPHT process. This argument is supported by XRD refinement results (Supplementary Fig. 8). If the transformation was only driven by shear, the BCC β -Li phase would be expected to transform into an HCP-structured Li phase. However, no corresponding diffraction signals for an HCP-Li phase were detected in XRD patterns. Furthermore, according to supplementary first-principles calculations (Supplementary Fig. 12b), the free energies of the BCC and HCP phases intersect at a Li concentration of approximately 23 at.%. This implies that when the local Li concentration exceeds this critical value, the BCC phase is more stable. Conversely, when the concentration falls below this value, the system spontaneously transforms from BCC to HCP via a martensitic transformation. Meanwhile, we have quantified the Li concentration changes within two phases during the MD simulation process (Supplementary Fig. 12a). Moreover, the reconstructed APT results indicated that the average Li concentrations are 3.3 at.% for the primary α -Mg phase and 20.2 at.% for the β -Li phase in the as-cast sample (Supplementary Fig. 15). In contrast, the analysis of distinct regions reveals an average Li concentration of 7.3 at.% for the Li-poor HCP variants and 14.5 at.% for Li-rich HCP

variants in the HPHT-800 sample (Supplementary Fig. 16). Both experimental and simulation evidence further supports that the martensitic-like transformation is primarily driven by Li compositional fluctuations.

During the HPHT process, high temperature provides a significant driving force for Li diffusion, while high pressure likely further promotes Li redistribution by affecting the chemical potential gradient. This leads to compositional fluctuations within the initial β -Li phase, creating local Li-poor regions (< 23 at.%). Once this thermodynamic condition is met, these regions undergo a martensitic transformation, generating HCP variants. This process can be supported by the updated TEM data of the HPHT-500 sample (Supplementary Fig. 14). The primary α -Mg regions and Li-rich regions are clearly distinguished by the HAADF-STEM image (the characteristic is consistent with the as-cast sample, Supplementary Fig. 13). A compositional diffusion layer exists between them, with no crystallographic difference observed (Supplementary Fig. 14b-e). More importantly, HCP variants with different orientations have begun to form within the Li-rich regions (Supplementary Fig. 13g-h). As the consumption of primary α -Mg regions, the HCP variants originating from different Li-rich regions interact with each other, ultimately forming the complex TIT network (Supplementary Fig. 14i). This direct observation provides key evidence for the continuous evolution process: "compositional fluctuation \rightarrow local phase transformation \rightarrow nucleation of multiple HCP variants \rightarrow formation of the TIT network." Overall, the TIT structure we discovered exists in the HCP structure produced by the martensitic transformation, rather than in the HCP region that has not undergone phase transformation. The significance of the dual-phase alloy design lies in the fact that the primary α -Mg lamellae serve as diffusion sites for Li. This leads to the formation of Li-poor regions within the primary β -Li phase, thereby inducing the martensitic transformation.

Supplementary Figure 1 | Microstructure of Mg-9Li samples. **a**, Surface morphology and dimension of HPHT samples. The diameter and height are 30 mm and 15 mm, respectively. **b-c**, IPF image and the distribution of β -Li phase (blue) of the as-cast sample. **d-e**, Optical structure, IPF image, and pole figures of the HPHT-800 sample. **f**, Grain size variation of the different HPHT Mg-9Li samples.

Supplementary Figure 12 | Martensitic transformation conditions. **a**, Li concentration variation in α -Mg and β -Li phases during MD simulation process. **b**, The variation curve between the energies of HCP and BCC structures dependent on the concentration of Li. The critical Li concentration is 23 at%.

Supplementary Figure 13 | Microstructure of as-cast Mg-9Li alloy. **a**, SEM image of the interface of primary α -Mg and β -Li in as-cast Mg-9Li alloy. **b**, HAADF-STEM image of the interface structure. **c**, EDS mapping of the β -Li region. **d**, BF-TEM image of the interface structure. **e**, HRTEM image of the interface structure viewed along $[0001]_{\alpha}$. **f**, HRTEM image of the interface structure viewed along $[11\bar{2}0]_{\alpha}$.

Supplementary Figure 14 | Microstructure of HPHT-500 Mg-9Li alloy. **a**, SEM image of the HPHT-500 Mg-9Li alloy. **b**, BF-TEM image of the primary α -Mg region viewed along $[1\bar{2}1\bar{3}]$ axis. **c**, HAADF-STEM image corresponding to the dashed box in **b**. **d**, High-magnification HAADF-STEM image of the diffusion layer. The inset shows the corresponding EDS mapping. **e**, HRTEM image of the diffusion layer viewed along $[1\bar{2}1\bar{3}]$. **f**, BF-TEM image of the Li-rich layer viewed along $[11\bar{2}0]$ axis. **g**, Local magnification image of **f**. **h**, BF-TEM image of the Li-rich layer viewed along $[11\bar{2}0]$ axis of the other HCP variant. **i**, High-magnification BF-TEM image of the interaction region between two Li-rich layers.

Supplementary Figure 15 | APT analysis of as-cast sample. a1-a3, Reconstructed APT volume showing the distribution of Mg and Li atoms. **b1-b2**, APT results showing the distribution of Li atoms defined by 10 at.% and 23 at.% Li iso-surfaces, respectively. **c**, Concentration profiles along the red arrow in **b1**. **d-e**, Concentration profiles corresponding to 10 at.% and 23 at.% Li iso-surfaces, respectively.

Supplementary Fig. 16 | APT analysis of HPHT-800 sample. a1-a3, Reconstructed APT volume showing the distribution of Mg and Li atoms. **b1-b3**, APT results showing the distribution of Li atoms defined by 10 at.% and 23 at.% Li iso-surfaces, respectively. **c-d**, Concentration profiles corresponding to 10 at.% and 23 at.% Li iso-surfaces, respectively.

3. Fig 2a: please increase its size or crop part of the image to enlarge the view to enable the visual identification of the TIT structures by the reader. Can you also provide the related diffraction pattern?

Response: Thank you for your valuable suggestion. In response to comments from other reviewers, we have re-imaged the key regions using aberration-corrected scanning transmission electron microscopy (HAADF-STEM). The relevant data have been updated in Fig. 2. Specifically, Fig. 2a has been replaced with a high magnification BF-STEM image, and the original Fig. 2a has been moved to Supplementary Fig. 6. In addition, we have updated the SAED patterns of two TIT structures (Supplementary Fig. 3).

Supplementary Figure 3 | SAED patterns. The SAED patterns of i-TIT (a) and a-TIT (b) structures viewed along [11 $\bar{2}$ 0], respectively.

Fig. 2 | Microstructures. **a**, BF-STEM image of TIT structures in an HPHT-800 Mg-9Li alloy viewed along $[11\bar{2}0]$. **b**, HAADF-STEM image of a typical i-TIT structure. **c**, HAADF-STEM image of a typical a-TIT structure. The yellow dashed lines and red dashed lines represent $\{10\bar{1}1\}$ TBs and $\{10\bar{1}3\}$ TBs, respectively. **d**, HAADF-STEM image of a typical $\{10\bar{1}1\}$ TB. Li atoms are arranged periodically on the $\{10\bar{1}1\}$ TB. **e-f**, the disclination structures of i-TIT and a-TIT, respectively. The blue solid line represents the Mg matrix plane, and the red solid line represents the $\{10\bar{1}1\}$ TBs. **g**, HRTEM images of basal intrinsic I1-type stacking faults around $\{10\bar{1}1\}$ TBs viewed along $[11\bar{2}0]$. **h**, The statistical fractions, determined on the basis of TEM images, of various interfaces.

4. Can you provide more details about the twin thickness measurement? How many grains were considered? Are they statistically relevant of the whole microstructure?

Response: Thank you for your advice. Due to the small size of twins, even at an acceleration voltage of 20-30 kV, the lateral diffusion scale of the electron beam spot in the Mg alloy is usually in the range of 50-100 nm. This means that the Kikuchi diffraction signal collected at each point actually represents a mixed signal from the twin lamella and the matrix on both sides. This results in degraded Kikuchi band quality, reduced indexing rates, and difficulty in clearly resolving the orientation difference between such thin twins and the matrix. Consequently, reliable and automated statistical identification of nanotwins over large areas is not feasible. Therefore, in this study, the statistics of the thickness of nanotwins must be performed manually using TEM images with higher spatial resolution.

"Twin thickness" is defined as the shortest perpendicular distance between two parallel twin boundaries. Our measurements are primarily based on multiple high-resolution TEM images taken from different regions of the HPHT-800 sample. We systematically measured twin thickness across three randomly selected grains. The TEM samples were prepared using focused ion beam (FIB) technology. In these samples, each region was oriented as close as possible to the [11-20] axis for twin thickness measurement. In total, over 100 measurement points were collected to support the main conclusions of this study. We believe that these data provide statistically meaning for the overall understanding of the microstructure. Additionally, we have updated EBSD data of the HPHT-800 sample (Supplementary Fig. 1). In indeed, these images are rather ambiguous for the clear identification of nanotwins.

Supplementary Figure 1 | Microstructure of Mg-9Li samples. **a**, Surface morphology and dimension of HPHT samples. The diameter and height are 30 mm and 15 mm, respectively. **b-c**, IPF image and the distribution of β -Li phase (blue) of the as-cast sample. **d-e**, Optical structure, IPF image, and pole figures of the HPHT-800 sample. **f**, Grain size variation of the different HPHT Mg-9Li samples.

5. Does the crystallographic orientation of the grains affect the TIT formation? on fig 2a, some grains seem to not contain TIT network. Can you also comment on the role of the misorientation between grains in that process?

Response: Thank you very much for your constructive suggestions. First, we would like to emphasize that the formation of the TIT structure in this study is not directly driven by external mechanical deformation. Rather, it originates from a martensitic transformation process triggered by local compositional fluctuations of Li atoms during the HPHT treatment. This process can be supported by the updated TEM data of the HPHT-500 sample (Supplementary Fig. 14). The primary α -Mg regions and Li-rich regions are clearly distinguished by the HAADF-STEM image (the characteristic is consistent with the as-cast sample, Supplementary Fig. 13). A compositional diffusion layer exists between them, with no crystallographic difference observed (Supplementary Fig. 14b-e). More importantly, HCP variants with different orientations have begun to form within the Li-rich regions (Supplementary Fig. 14g-h). As the

consumption of primary α -Mg regions, the HCP variants originating from different Li-rich regions interact with each other, ultimately forming the complex TIT network (Supplementary Fig. 14i). This direct observation provides key evidence for the continuous evolution process: "compositional fluctuation \rightarrow local phase transformation \rightarrow nucleation of multiple HCP variants \rightarrow formation of the TIT network."

Moreover, we attempt atomic-scale characterization of the phase interface using aberration-corrected scanning transmission electron microscopy (STEM). However, fundamental limitations arise due to the extremely low atomic number of Li ($Z=3$). Therefore, we employ MD simulations as a crucial supplementary approach. The simulations accurately reproduced the martensitic transformation process and have directly extracted the crystallographic orientation of the transformed regions (Supplementary Fig. 11). The results show that the orientation relationship between the newly formed HCP phase and the parent BCC phase is consistent with the classical martensitic transformation crystallography proposed in our manuscript. In summary, the crystallographic orientation of the grains makes a very small contribution to the formation of the TIT network.

Regarding your claim that some grains do not display the TIT network, this primarily results from differences in grain orientation that prevent the twins from being visualized, rather than the actual absence of twin structures in those regions. To address your question, we provided the following TEM data for clarification. Specifically, when capturing Fig. a, we tilted the grain with a clearly visible TIT network to the $[11-20]$ axis to present the $\{10-11\}$ twins. However, neighboring grains (appearing as regions with different contrast in Response Supplementary Fig. 6a) possess distinctly different crystallographic orientations. Due to the limited tilt range of the TEM double-tilt holder (typically $\pm 25^\circ$ to $\pm 30^\circ$), it is not possible to tilt both grains with such a large misorientation simultaneously to their respective $[11-20]$ axes within the same field of view. Consequently, regions that do not show TIT contrast under a specific imaging condition are not inherently "twin-free." Instead, their orientation deviates from the optimal imaging condition, rendering the twin contrast invisible. By

tilting and analyzing adjacent grain under different orientations, we have obtained sufficient data demonstrating that these regions also contain twin structures (Response Supplementary Fig. 6b-d).

Response Supplementary Figure 6 | a, BF-TEM image of HPHT-800 sample viewed along $[112\bar{0}]$ direction. b, BF-TEM image of the region marked by the red dashed box viewed along $[112\bar{3}]$ direction. c, High-resolution BF-TEM image of the red dashed box marked in b. d, HRTEM image corresponding to the nanotwin marked by the red arrow in c.

Supplementary Figure 11 | Crystallographic relationship of Martensitic transformation. Tensile deformation was performed along the $[1\bar{1}1]_{\text{BCC}}$ direction at 0 K. The blue region represents α -Mg with an HCP structure, the red region represents β -Li with a BCC structure, and the yellow region represents α -Mg that contains SF structure. **a**, BCC structure in Mg-9Li supercell viewed along $[1\bar{1}1]_{\text{BCC}}$. HCP phase begins to nucleate when the strain reaches 5 %. Subsequently, HCP variants with different orientations gradually interact and form twin structures as the strain increased to 6%. Finally, complete TIT structure has been formed as the strain increased to 10%. **b**, The same results also have been viewed along $[113]_{\text{BCC}}$ direction. **c**, The crystallographic relationship of martensitic transformation viewed along different directions.

Supplementary Figure 13 | Microstructure of as-cast Mg-9Li alloy. **a**, SEM image of the interface of primary α -Mg and β -Li in as-cast Mg-9Li alloy. **b**, HAADF-STEM image of the interface structure. **c**, EDS mapping of the β -Li region. **d**, BF-TEM image of the interface structure. **e**, HRTEM image of the interface structure viewed along $[0001]_{\alpha}$. **f**, HRTEM image of the interface structure viewed along $[11\bar{2}0]_{\alpha}$.

Supplementary Figure 14 | Microstructure of HPHT-500 Mg-9Li alloy. **a**, SEM image of the HPHT-500 Mg-9Li alloy. **b**, BF-TEM image of the primary α -Mg region viewed along $[1\bar{2}1\bar{3}]$ axis. **c**, HAADF-STEM image corresponding to the dashed box in **b**. **d**, High-magnification HAADF-STEM image of the diffusion layer. The inset shows the corresponding EDS mapping. **e**, HRTEM image of the diffusion layer viewed along $[1\bar{2}1\bar{3}]$. **f**, BF-TEM image of the Li-rich layer viewed along $[11\bar{2}0]$ axis. **g**, Local magnification image of **f**. **h**, BF-TEM image of the Li-rich layer viewed along $[11\bar{2}0]$ axis of the other HCP variant. **i**, High-magnification BF-TEM image of the interaction region between two Li-rich layers.

6. What is the percentage of i-TIT and a-TIT in your material? Can it be controlled somehow? Does it affect the overall plastic mechanisms and mechanical properties of the alloy?

Reponses: Thank you very much for your question. At this research stage, the primary objective was to reveal and elucidate the crucial role of this TIT structure as a novel strengthen-ductility carrier in Mg-Li alloys. Therefore, the focus has been on qualitatively confirming its existence, analyzing its formation mechanism, and quantitatively evaluating its macroscopic mechanical effects. Based on current experimental observations, the volume fractions of i-TIT and a-TIT structures in the material are approximately 42.85% and 57.15%, respectively (close to 1:1) (Supplementary Fig. 6). We acknowledge that an accurate connection between processing parameters and microstructure has not been established, which is indeed a complex and fascinating challenge. Precisely controlling the ratio of the two TIT structures is undoubtedly the next key goal for enabling customized microstructural design and subsequent performance optimization. This will be a central focus of our future research efforts.

Supplementary Figure 6 | Homogeneous distribution characteristics of TITs. **a**, BF-TEM image of TIT structures in an HPHT-800 Mg-9Li alloy viewed along $[11\bar{2}0]$. **b-m**, HRTEM images taken from the randomly selected fields of view within the red square regions in **a**. The viewing direction is parallel to $[11\bar{2}0]\alpha$. The yellow dashed lines, red dashed lines, and blue dashed lines represent $\{10\bar{1}\}$ TBs, $\{10\bar{1}3\}$ TBs, and BP facets, respectively.

7. Lines 109 to 111: the statement about the density of the structure is not clear. Can you quantify the different densities measured and explain how they were measured?

Response: Thanks for the very careful review. We intended to indicate that the two types of TIT structures differ in their numerical density. To avoid any misunderstanding by readers, we have replaced the term "density" with "fraction" in the relevant description.

We have made the following modifications to the manuscript:

“In detail, a representative ellipsoid-shaped characteristic of the TIT structure is first detected (Fig. 3d-e), in which the fractions of i-TIT and a-TIT structures are approximately 42.85% and 57.15%, respectively (Supplementary Fig. 6).”

8. Figure 2k: a GB proportion is mentioned while no GB is visible on the 12 analyzed structures. Can you clarify that?

Response: Thank you very much for your question. “Additionally, we analyzed twelve randomly distributed interface structures (Supplementary Fig. 5) and observed that the TIT structures were uniformly distributed in the HPHT-800 Mg-9Li alloy. The interfaces were composed mainly of original GBs, TBs and basal-prismatic (BP) interfaces, and the average interface fraction of TBs was approximately 65.7% (Fig. 2k).” We believe your confusion stems from this sentence. This is a misunderstanding caused by our ambiguous description when writing the manuscript. Our statistics on the interface fraction are not based on those 12 structures, but rather a large-scale statistic of the entire sample (for example, Supplementary Fig. 2a and Response Supplementary Fig. 7).

To avoid any misunderstanding, we have revised this sentence to follows:

“Additionally, we analyzed randomly distributed interface structures (Supplementary Fig. 6) and observed that the TIT structures were uniformly distributed in the HPHT-800 Mg-9Li alloy. The interface structures were composed mainly of original GBs, TBs

and basal-prismatic (BP) interfaces, and the average interface fraction of TBs was approximately 65.7% (Fig. 2h).”

Response Supplementary Figure 7 | BF-TEM image of HPHT-800 sample viewed along $[11\bar{2}0]$ direction.

9. Please clarify the role of stacking faults on the TIT formation. Why do MD simulations not emphasize the role of stacking faults?

Response: Thank you very much for your constructive suggestions. To clarify the role of stacking faults on the TIT formation, the generalized planar fault energy curve was calculated using first-principles methods (Response Supplementary Fig. 8). A BCC Mg without stacking faults was taken as the reference state. The formation of a stacking fault requires overcoming an energy barrier of 0.18 J/m^2 . Interestingly, when a stacking fault is already present in the BCC Mg, only 0.13 J/m^2 is needed to form a second stacking fault. This result highlights that stacking faults can promote martensitic transformation, specifically the formation of TIT. Indeed, in our MD simulations, numerous stacking faults were observed during the phase transformation and within the transformed HCP Mg, as shown in Fig. 4. In our manuscript, we primarily focused on the BCC to HCP phase transformation and the accompanying changes in Li concentration. Therefore, the role of SFs have not been clarified.

Response Supplementary Figure 8 | The generalized planar fault energy curve of BCC Mg shear along the [111] direction.

10. You present hardness variation over 2h while comparing later the tensile properties of the alloys after 2 years of ageing. What is the corresponding hardness after 2 years of ageing ? Also how do you explain the drop of hardness when a temperature higher than 800 C is used for the second step of HPHT? Please also explain why no difference in hardness is observed after the first step of HPHT?

Response: Thank you very much for your constructive suggestions. We have included the complete natural aging hardness variation curve over two years. To address your question and allow readers to readily access the data, the hardness value after two years of aging has been annotated in Fig. 5d. Regarding the observed decrease in hardness, we have supplemented the response with an optical micrograph of the HPHT-1100 sample, which reveals coarse twin lamellae within the grains (Response Supplementary Fig. 9). Therefore, the decline in hardness for samples treated at temperatures above 800°C during HPHT processing is primarily attributed to the coarsening of the nanotwins. The absence of a significant hardness increase after the first HPHT step is due to the insufficient pressure and temperature to induce the martensitic transformation, resulting in negligible change in hardness.

Fig. 5 | Mechanical properties and deformed characteristics. Unprecedented stability of the HPHT-800 Mg-9Li alloy in comparison with that of other reported Mg-Li alloys at ambient temperature.

Response Supplementary Figure 9 | Optical image of the HPHT-1100 sample.

11. Fig 4a: the stress is not supposed to decrease in a true-stress/true-strain graph. Can you please remove the data obtained after strain localization arises in the tensile specimens? Can you also detail how the yield stress was calculated?

Response: Thank you for pointing out the problem. The authors agree that the stress in the true stress-strain curve should not decrease during the plastic deformation stage. The drop might be due to invalid data after necking or data processing errors. Accordingly, we have re-tested the mechanical properties of the HPHT-900 sample and updated the corresponding results in Fig. 5a. Regarding the method you mentioned for calculating the yield stress, we draw a line parallel to the linear part of the elastic stage on the strain axis, offset by 0.2% from the origin, and the intersection point of this line with the stress-strain curve is the yield stress.

Fig. 5 | Mechanical properties. Tensile true stress–strain curves of Mg-9Li alloys in various states at a nominal strain rate of 10^{-3} s^{-1} .

12. The strain hardening observed on the specimen deformed after 2 years of ageing seems higher than the hardening observed on a “fresh” specimen. Can you comment? How are affected the TIT-dislocation interaction mechanisms?

Response: We would appreciate your valuable comments. In our study of the TIT structure, we selected the binary Mg-Li alloy as a model material. This system does not contain metastable precipitates, which are commonly found in ternary systems such as Mg-Li-Zn or Mg-Li-Al alloys. In HPHT-800 Mg-9Li alloy, the slight hardness degradation is mainly attributed to the thickening of $\{10\text{-}11\}$ nanotwins and the relaxation of internal stresses (Supplementary Fig. 21). The twin thickness plays a critical role in influencing strain hardening.

Firstly, when twins are relatively thin, the limited space within the twin lamellae restricts dislocation activity, storage, and interaction, which can lead to reduced or saturated work-hardening capacity. Secondly, when twins thicken to a certain extent, sufficient space becomes available within the lamellae for dislocation slip, multiplication, and storage. This can generate strong back-stress hardening and forest dislocation hardening, thereby resulting in higher strain hardening.

Supplementary Figure 21 | Long-term stability of TIT structures. **a**, BF-TEM image of the HPHT-800 Mg-9Li sample aged at ambient after two years. Based on statistical analysis, the average thickness of the nanotwin is 67 nm. **b-c**, HRTEM images of the i-TIT and the a-TIT structures in white dashed box indicated in **a**, respectively. **d**, BF-TEM image of the HPHT-800 Mg-9Li sample aged at 60 °C after 2000 h. Based on statistical analysis, the average thickness of the nanotwin is 78 nm. **e-f**, HRTEM images of the i-TIT and the a-TIT structures in white dashed box indicated in **d**, respectively.

13. Why do your MD simulations not predict the role of FCC structures during the deformation process? To what extent is the microstructure used to model the alloy comparable to the experimental microstructure observed? Can you explain the modelling strategy used to generate the initial microstructure?

Response: Thank you for your beneficial suggestion. In our MD simulations (Fig. 3d), a substantial number of stacking faults arranged in a layer-by-layer can be observed, beyond isolated stacking faults. This structure actually corresponds to a distorted area with nearly FCC structure. Therefore, we believe you may have some misunderstandings about our simulations, which may stem from the description of “the yellow region represents α -Mg that contains a TB/SF structure”. To eliminate this misunderstanding, the related statements have been revised.

A large-scale supercell with dimensions of $7.9 \times 44.1 \times 46.1 \text{ nm}^3$ was constructed for our MD simulations, in which the BCC Li-rich phase and HCP Mg-rich phase are connected by a coherent interface. Since grain boundaries were not included in the simulation setup, the present results offer only an atomic-scale view of the BCC-HCP phase transformation, without capturing grain-boundary-related microstructures.

14. Insufficient information is provided on nanopillar compression experiments. Do they contain GB? Does the sample preparation affect the TIT structures? What is the crystallographic orientation of the pillar?

Response: Thanks for the very careful review. Our nanopillars were extracted from grain interiors, with a diameter of only 300 nm, and therefore do not involve any GB effects. Additionally, the nanopillars were prepared using the FIB annular milling technique, and the TEM data were also obtained from FIB-fabricated samples. Consequently, the influence of the sample preparation process on the TIT structure is negligible.

When conducting the compression experiment of the nanopillars, we did not characterize its crystallographic orientation. However, to eliminate this influence,

we took samples from random grains for testing respectively. Furthermore, the updated EBSD data show that the HPHT-800 sample exhibits a random crystallographic orientation (Supplementary Fig. 1). Therefore, we have reason to believe that the two compression curves with similar yield strengths are sufficient to demonstrate the isotropic mechanical contribution of the TIT structure.

Supplementary Figure 1 | Microstructure of Mg-9Li samples. a, Surface morphology and dimension of HPHT samples. The diameter and height are 30 mm and 15 mm, respectively. b-c, IPF image and the distribution of β -Li phase (blue) of the as-cast sample. d-e, Optical structure, IPF image, and pole figures of the HPHT-800 sample. f, Grain size variation of the different HPHT Mg-9Li samples.